# Field monitoring of pore-water pressure in fully and partly saturated debris flows at Ohya landslide scar, Japan

Syunsuke Oya[1],Fumitoshi Imaizumi[2], Shoki Takayama[2]

[1]Nippon Koei Co., Ltd., Fukuoka, 812-0007, Japan
[2]Faculty of Agriculture, Shizuoka University, Shizuoka, 422-8429, Japan

*Correspondence to*: Fumitoshi Imaizumi (imaizumi@shizuoka.ac.jp)

**Abstract.** The characteristics of debris flows (e.g., mobility, sediment concentration, erosion, and deposition of sediment) are dependent on the pore-water pressure in the flows. Therefore, understanding the magnitude of pore-water pressure in debris flows is essential for improving debris flow mitigations measures. Notably, the pore-water pressure in a partly saturated flow,
which contains an unsaturated layer in its upper part, has not been understood, due to a lack of data. The monitoring performed in Ohya landslide scar, central Japan, allowed us to obtain the data on the pore-water pressure in fully and partly saturated flows during four debris flow events. In some partly and fully saturated debris flows, the pore-water pressure at the channel bed exceeded the hydrostatic pressure of clean water. The depth gradient of the pore-water pressure in the lower part of the flow, monitored using water pressure sensors at multiple depths, was generally higher than the depth-averaged gradient of the pore-
water pressure from the channel bed to the surface of the flow. The low gradient of the pore-water pressure in the upper part of partly saturated debris flow may be affected by the low hydrostatic pressure due to unsaturation of the flow. Bagnold number, Savage number, and Friction number indicated that frictional force dominated in the partly saturated debris flows. Excess pore-water pressure was observed in the lower part of partly saturated surges. The excess pore-water pressure may have generated by the contraction of interstitial water and have maintained due to low hydraulic diffusivity in debris flows. The pore-water pressure
at the channel bed of fully saturated flow was generally similar to the hydrostatic pressure of clean water, while some saturated surges portrayed higher pore-water pressure than the hydrostatic pressure. The travel distance of debris flows, investigated by the structure from motion technique using unmanned aerial vehicle (UAV-SfM) and the monitoring of time lapse cameras, was long during a rainfall event having high intensity, even though the pore-water pressure in the flow was not significantly high. We conclude that the excess pore-water pressure is present in many debris flow surges and an important mechanism in the debris
flow surge behaviors.

## 1 Introduction

Debris flows are hydrogeomorphic processes in steep mountain channels that can cause severe damages to property and life, due to their high velocity, large volume, and destructive power (Scott et al., 2005; Cui et al., 2011; Kean et al., 2019). Debris flows with higher pore-water pressure have a higher flow mobility and longer travel distance because the pore-water pressure decreases
resistance in the flow by decreasing effective stress among boulders (de Boer and Ehlers, 1990; Iverson, 1997; Hotta, 2012). Therefore, analyzing the pore-water pressure in debris flows is important for determining the areas that need debris-flow hazard mitigation measures.

Debris flow observations have been conducted in several countries, such as the United States of America (Kean et al., 2011; McCoy et al., 2013), Switzerland (Berger et al., 2011; Walter et al., 2017), Italy (Arattano et al., 2012; Marchi et al., 2021),
China (Hu et al, 2011; Cui et al., 2018), and Japan (Okano et al., 2012; Osaka et al., 2014). However, a few observations have been conducted in the initiation zones of debris flows (McCoy et al., 2012; Kean et al., 2013; Simoni et al., 2020). Debris flows

are classified into various flow types, based on the particle size, rheology, and solid concentrations (Coussot and Meunier, 1996; Takahashi, 2014). Fully/partly saturated debris flows correspond to the saturation/unsaturation of interstitial water in the upper part of the flows (Imaizumi et al., 2005, 2019). Partly saturated flows, which have an unsaturated layer in their upper part, have been frequently observed in steep debris-flow initiation zones (Imaizumi et al., 2019). The formation of partly saturated flows in steep channels can be explained by analyzing the balance of static force (i.e., shear stress and shear resistance) at the bottom of the sediment mass (Imaizumi et al., 2017). However, partly saturated flows, which originate in the steep channels, continually migrate down to channel sections that have a gentle slope gradient (McArdell et al., 2007; McCoy et al., 2010; Okano et al., 2012). At these sections, static force analyses indicate that the shear resistance exceeds the shear stress (Imaizumi et al., 2017). Thus, the internal pressure and stress in partly saturated flows need to be determined, to explain why partly saturated flows have high mobility, without being rich in interstitial water.

Previous studies have conducted laboratory experiments to understand the pore-water pressure in debris flow (Major, 2000; Deangeli, 2009; Zhou et al., 2019). Notably, pore pressure consists of hydrostatic and excess pore-water pressures (Hampton, 1979). Hydrostatic pressure in the debris flow can occasionally be higher than that in clear water, because of the higher weight per unit volume of interstitial water in the debris flow, due to the inclusion of suspended fine particles (Iverson, 1997; Kaitna et al., 2016). Numerical simulations indicate that an increase in the suspended fine particles result in a longer travel distance of debris flow (Uchida et al., 2020). The excess pore-water pressure occurs due to the contraction of interstitial water by the surrounding boulders in a high shear stress environment (Iverson, 1997, 2005; Iverson and George, 2014; Kaitna et al., 2016), Reynolds stress from the turbulence of interstitial water caused by the collision of boulders (Zenit and Hunt, 1998; Hotta and Ohta, 2000; Hotta, 2011), and centrifugal force in curved channel sections (Hotta, 2012). The generation and preservation of excess pore-water pressure increases debris flow mobility (Iverson and Vallance, 2001; Lanzoni et al., 2017). Note that debris flows with rich fine particles can preserve excess pore-water pressure for longer periods, owing to the low hydraulic conductivity in the flow (Major and Iverson, 1999; Okada and Ochiai, 2008; de Haas et al., 2015). The grain size distribution of large particles in the debris flow also affects the preservation of excess pore-water pressure (Bowman and Sanvitale, 2009; Kaitna et al., 2016; Yang et al., 2021a). Previous studies indicate that the debris flow simulations that consider excess pore-water pressure can better portray the real areas affected by debris flow than those that ignore the excess pore-water pressure (Abe and Konagai, 2016; Pastor et al., 2021).

Although laboratory experiments can provide details of the stresses and pressures in debris flows, it is difficult to accurately reproduce the stresses and pressures in real debris flows. For example, most laboratory experiments underestimate the effect of excess pore-water pressure in the debris flow mobility, due to the small scales of the experiments (Iverson, 1997). Field observations of pore-water pressure have also conducted in active debris flow torrents (Berti et al., 2000; McArdell et al., 2007; McCoy et al., 2010; Nagl et al., 2020). In general, debris flow surges with high excess pore pressure travel longer distances (McCoy et al., 2010). Excess pore-water pressure can be present or absent in different debris flows, even if they belong to the same torrent (Nagl et al., 2020). In previous studies, pore-water pressure was mainly observed in the main and subsequent flows of surges (Berti et al., 2000; McArdell et al., 2007; McCoy et al., 2010; Nagl et al., 2020), and only a few present the monitoring data on the pore-water pressure in partly saturated flow at the front of surges Therefore, the mechanisms of the migration of partly saturated debris flows remains unclear.

For every debris flow event, the flow characteristics (e.g., travel distance, velocity, and sediment concentration), which are affected by rainfall pattern and the volume of channel deposits in debris flow initiation zone, vary significantly (Hürlimann et al., 2003; Okano et al., 2012). The flow characteristics of different surges, even in the same debris flow event, can vary significantly (Theule et al., 2018; Itoh et al., 2021). These results imply that the magnitude of excess pore-water pressure in different events

and surges can vary. Nevertheless, a common understanding of the variations in the pore-water pressure remains unclear and limited.

In the Ichinosawa catchment within the Ohya landslide scar, central Japan, intensive field monitoring has been carried out since 1998 (Imaizumi et al., 2005, 2006). Debris flows, which occur due to the mobilization of storage around channels (i.e., talus cone and channel deposits), occurs frequently (about three or four events per year), because of active dry ravel and rockfall from outcrops during freeze-thaw periods. Both fully and partly saturated debris flows occur in the Ichinosawa catchment, because of the steep terrain of the region (Imaizumi et al., 2017, 2019).

The aim of this study is to understand the characteristics of pore-water pressure in partly and fully saturated debris flows. The depth profile of the pore-water pressure in debris flows in Ichinosawa catchment was monitored using water pressure sensors. The specific objectives of the study were to (1) clarify the depth gradient of pore-water pressure; (2) reveal the factors that affect the magnitude of the pore-water pressure; and (3) discuss the influence of the pore-water pressure on the runout characteristics of debris flows.

## 2 Study area

The Ohya landslide, which had a total volume of 120 million $m^3$, was initiated during an earthquake in 1707 CE (Tsuchiya and Imaizumi, 2010). The outcropping bedrock was composed of well-jointed sandstone and highly fractured Paleogene shale. The annual precipitation at the site is about 3,400 mm (Imaizumi et al., 2005). Heavy rainfall (i.e., total rainfall >100 mm) occurs during the Meiyu-Baiu rainy season (from June to July) and the autumn typhoon season (from August to October).

Most of debris flow in the Ohya landslide occurs in the Ichinosawa torrent, which flows down from the north to the center of the landslide (Fig. 1a). The Ichinosawa catchment has an altitude of 1,270–1,905 m above sea level (m a.s.l.), with an area of 0.3 $km^2$ and channel length of ca. 1,000 m. The Ichinosawa catchment can be divided into two sections, upper and lower Ichinosawa, separated by a waterfall named "Ohya-ohtaki" (altitude: 1450 m a.s.l.) (P20 in Fig. 1a). The upper Ichinosawa region, which is the initiation zone of debris flows in the area, is characterized by a deeply incised channel and steep slopes (40–65°; Fig. 1b). Notably, 70 % of the slope is scree and outcropping bedrock, and the remaining 30 % is covered with vegetation (trees, shrubs, and tussocks). Rockfall and dry ravel promoted by the freeze-thaw process in winter and early spring is the predominant sediment infilling process (Imaizumi et al., 2006). A large volume of sediment, ranging from sand particles to boulders, is stored in the channel bed and talus cones (Imaizumi et al., 2006; Imaizumi et al., 2017). The channel gradient is generally steeper than 25°, and in the uppermost part, the channel gradient is close to the talus slope (37.3°) (Fig. 1d; Imaizumi et al., 2017). The lower Ichinosawa is a debris flow fan (Fig. 1c). The channel gradient in the lower Ichinosawa is mainly 15–20° (Fig. 1d). The depth of deposits in the debris flow fan was estimated to be at least 5 m, based on previous channel bed changes interpreted visually by periodic field surveys conducted since 1998.

The Ichinosawa torrent joins the Hontani torrent at 1300 m a.s.l., at the center of Ohya landslide (Fig. 1a), merging into the Ohya River. Some debris flows pass through the junction with the Hontani torrent (e.g., one debris flow every several years) and flow down the Ohya River (Imaizumi et al., 2016b).

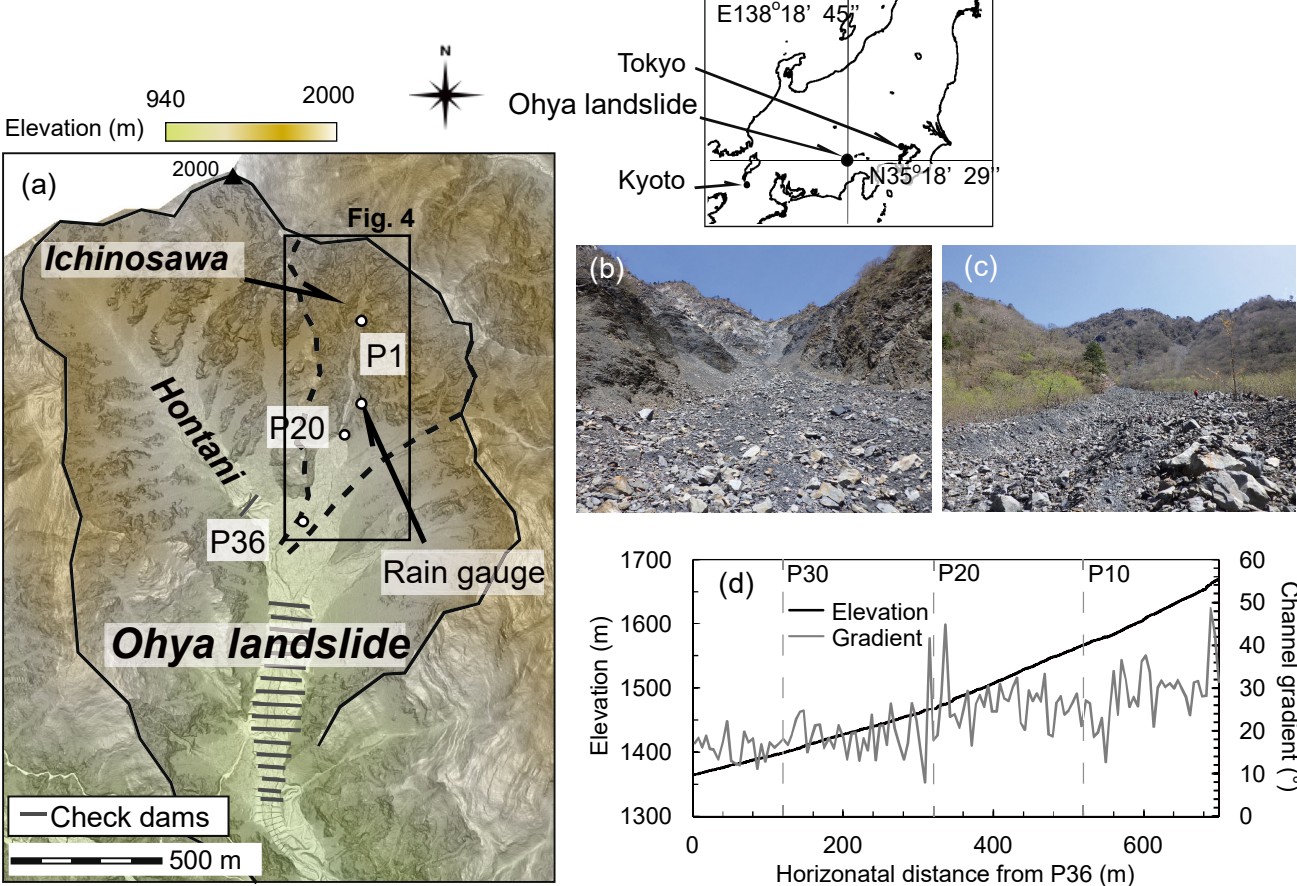

**Figure 1. Map and photographs of the Ohya landslide. (a) Map of the Ohya landslide. Airborne light detection and ranging (LiDAR) digital elevation model (DEM) provided by Ministry of Land, Infrastructure, Transport and Tourism (MLIT), Japan, was used to draw the map. P1 to P36 are analysis points of time lapse camera (TLC) images. Locations of all analysis points are indicated in Fig. 4. (b) Photograph of channel deposits around the rain gauge. (c) Debris flow deposits at P36 on the debris flow fan. The photographs were captured on April 25, 2022. (d) Graph of the longitudinal profile and channel gradient from P36 to P1 along the debris flow torrent; the channel gradient was calculated at intervals of 5 m on July 3, 2020, using DEM.**

## 3 Methodology

The debris flows in the Ohya landslide scar in central Japan have been monitored since 1998 (Fig. 1). The monitoring system consists of time lapse cameras, rain gauges, and water pressure sensors. The pore-water pressure was monitored at multiple depths of the channel from April 2020 to October 2021. A video camera and a laser distance sensor were installed in 2021. The observations were conducted from April to October; the observations from November to March were intermittent, to avoid the damages to the system from rockfall and snow avalanche. Periodical photography of the torrent since 1998 reveal that few debris flows occur in the intermission seasons from November to March (Imaizumi et al., 2006).

### 3.1 Debris flow image

In April 2020, 27 time-lapse cameras (TLCs) (Brinno, TLC200Pro and TLC2000, Taiwan) were installed along the Ichinosawa torrent in the section between P1and P36 (Fig. 1). The TLCs captured the images of the runout characteristics of the debris flows from the initiation to the deposition zones. The image resolutions of the TLC200Pro and TLC2000 cameras were 1280 × 720 and 1920 × 1080 pixels, respectively. The intervals of images, which range from 1 s to 10 s, was different for different cameras and periods. Additionally, the number of TLC was reduced to 21 in April 2021. We considered the analysis points (P1 to P36) along

the main channel of the Ichinosawa catchment, with an interval of about 20 m, to interpret the arrival timing of the debris flow surges, using the TLC images. Some points could not be analyzed, because they were not covered by the TLC images.

A video camera (Sony, HDR-CX470, Japan), with a frame rate of 60 frames per second and a resolution of 1,920 × 1,080 pixels, was installed at P20 in May 2021, to capture the motion images of the debris flows. The video camera images were

initiated by a rope sensor that could detect the motion of boulders on the channel bed. This video camera successfully captured the debris flow that occurred on July 13, 2021 (see information of supplement at the end of this paper).

We visually identified the temporal changes in the flow type (partly and fully saturated flows) at the analysis points of the TLC images (P1 to P36), based on the existence of interstitial water on the flow surface (Imaizumi et al., 2017). The fully saturated flows were turbulent and characterized by a black surface, due to the high concentrations of silty sediments sourced

from the shale in the interstitial water that filled the matrix of boulders. In contrast, muddy water could not be identified in the flow surface matrix of partly saturated flows. In this study, we classified the debris flow surges in the study site into three types: surges just composed of partly saturated flows, surges composed of both partly and fully saturated flows, and surges composed of only fully saturated flows (Fig. 2).

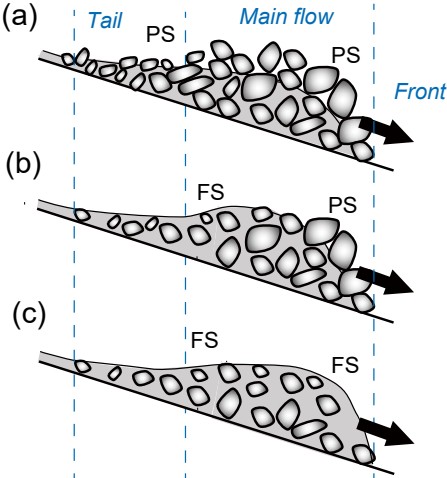

**Figure 2. Classification of the debris flow surges in the study area: (a) surges composed of only partly saturated flows, (b) surges composed of both partly and fully saturated flows, and (c) surges composed of only fully saturated flows. PS and FS indicate partly and fully saturated flows, respectively**

### 3.2 Pore-water pressure

Pore-water pressure sensors (KELLER AG für Druckmesstechnik, PR-26Y, Switzerland), which measure the pressure in the

range of 0–98,000 Pa, with accuracy of ±0.25 %, were installed at two heights (S1 and S2, at height of 0 m and 0.33 m from the channel bed, respectively) of the channel bank with exposed bedrock at P20 in May 2020. The sensors were installed in holes made by a hammer drill and were covered by mortar in order to reduce effect of the dynamic pore-water pressure caused by direct hit of the lateral flow (Fig. 3). The inlet of water into the sensors faces downward direction for the same reason. During most of the monitoring period, channel deposit accumulations were not observed at P20; however, the accumulations were

monitored for short periods after the deposition of sediments due to a few debris flow surges. Three additional pressure sensors were installed at 3 heights (S3, S4, and S5, at heights of 0.64, 0.91, and 1.17 m from the channel bed, respectively) along the same cross-sectional line in May 2021. In 2021, two sensors (S2 and S3 at heights of 0.33 m and 0.64 m, respectively) failed to observe the pore-water pressure of some debris flows, because of mechanical issues. A boulder on the channel bed and a data logger (Campbell scientific, CR200X, United States) were connected by a rope; the model generated electric signals when the

boulder was transported by debris flows. The data logger started operating (at 4-s intervals) when the electric signal was detected.

The monitored pressure was subtracted by the atmosphere pressure, which was monitored by an atmosphere pressure sensor (Oyo corporation, S&DL mini, Japan).

The components of the pore-water pressure, $p$, can be expressed as follows (McArdell et al., 2007; Kaitna et al., 2016):

$$p = p_h + p_e = p_w + p_s + p_e \qquad (1)$$

where $p_h$ is the hydrostatic pressure, $p_e$ is the excess pore-water pressure, $p_w$ is the static pressure of clean water, and $p_s$ is the static pressure from suspended fine sediments. The $p_e$ occurs due to the contraction of interstitial water by the surrounding boulders, Reynolds stress from the turbulence of interstitial water, and centrifugal force in curved channel sections (Iverson, 1997; Kaitna et al., 2016; Zenit and Hunt, 1998; Hotta, 2012). The hydrostatic pressure of clean water at the channel bed can be expressed by Eq. (2), as follows:

$$p_w = \rho g h \qquad (2)$$

where $\rho$ is the weight per unit volume of water (1000 kg m$^{-3}$), $g$ is the gravitational acceleration (9.8 m s$^{-1}$) and $h$ is the flow depth (m). The $\partial p / \partial z$ was 9,800 Pa m$^{-1}$ in the case of clean water, without any excess pore-water pressure ($p_s = p_e = 0$). By assuming that the weight per unit volume of sediment was 2,650 kg m$^{-3}$, we also calculated the depth gradient of normal stress in case that the space is completely filled by the sediment (25,970 Pa m$^{-1}$). This depth gradient can be considered as the maximum static pressure of interstitial fluid, because the interstitial fluid is mixture of fine sediments and water with lower weight per unit volume than the sediment. In other words, the excess pore-water pressure, $p_e$, certainly existed in the debris flows, if the $\partial p / \partial z$ exceeded 25,970 Pa m$^{-1}$. Both or one of $p_e$ and $p_s$ affected the pore-water pressure, when the $\partial p / \partial z$ was 9,800–25,970 Pa m$^{-1}$. In this study, $\Delta p / \Delta z$ obtained by two different method was used as a substitute for $\partial p / \partial z$. The first method provides the representative (depth-averaged) $\Delta p / \Delta z$ from the channel bed to the surface of flow, which can be expressed as follows:

$$\frac{\Delta p}{\Delta z}_{entire} = \frac{p_l}{h} \qquad (3)$$

where $p_l$ is the water pressure observed by the sensor at the channel bed (S1 in Fig. 3). The second method, shown in Eq. (4), was used to obtain the $\Delta p / \Delta z$ values for the lower part of the debris flow, using the monitoring data of the two pore-water pressure sensors at different depths.

$$\frac{\Delta p}{\Delta z}_{lower} = \frac{-(p_u - p_l)}{h_u - h_l} \qquad (4)$$

where $p_u$ is the water pressure observed by the upper sensor, and $h_u$ and $h_l$ are the heights of upper and lower sensors, respectively. The lower sensor in Eq. (4) was S1 in all periods. The upper sensor in Eq. (4) was the sensor at the lowest altitude above S1 that could be used to observe the pore-water pressure effectively.

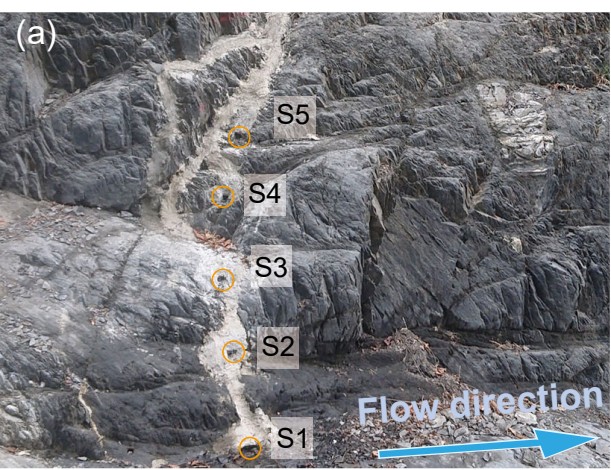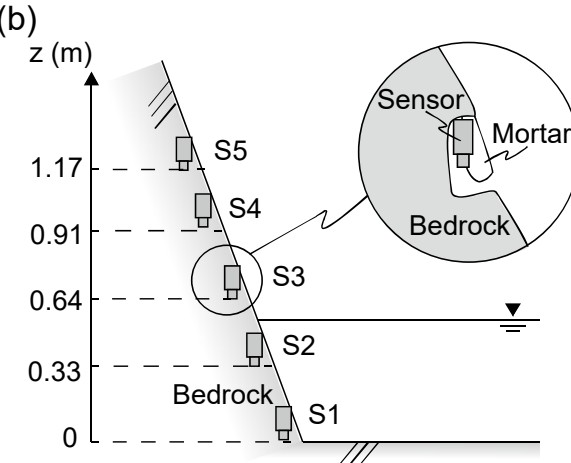

Figure 3. Installation of water pressure sensors at P20: (a) a photograph portraying the installation points of the water pressure sensors; (b) schematic diagram portraying the heights and protection mechanism of the sensors.

### 3.3 Flow depth

We obtained the flow depths above the water pressure sensors at the P20 site, based on the visual analysis of the images captured by the TLCs. The height of the flow surface was estimated by comparing the locations of the flow surfaces in the images to the scales of the surveying poles set at the same locations in the images that were captured on the days without debris flows. The flow depth was the distance from the channel bed with exposure of the bedrock to the height of flow surface. The flow depth was also monitored by a laser distance sensor (Sick, DT35, German), having an accuracy of 10 mm, placed above the water pressure sensors. The data logging of the laser distance sensors, having an interval of 4 s, was initiated by a rope sensor that could detect the motion of boulders on the channel bed.

### 3.4 Precipitation

Precipitation was measured for a logging interval of 1 min, using a tipping bucket rain gauge (0.2 mm for one tip) located in an open area near the P14 site (Fig. 1b). The duration used to separate different rainfall events (inter-event time definition, IETD) was set to 6 h, which could nicely determine rainfall threshold of debris flows (Imaizumi et al., 2017). The total rainfall depth and 10-min rainfall intensity was calculated based on a 1-min time step. We also calculated the cumulative rainfall (from the beginning of rainfall to the peak rainfall intensity) that triggered the debris flow surges in the study site. In August 2021, we could not record the precipitation, due to a mechanical issue with the equipment. For this period, we used the precipitation data observed by the Ministry of Land, Infrastructure, Transportation, and Tourism (MLIT) in Japan. The rain gauge of the MLIT (0.5 mm for one tip) was located 200 m south from the lower end of the Ichinosawa catchment.

### 3.5 Volume of sediment storage

The temporal changes in the surface topography of the sediment storage in the Ichinosawa associated with the occurrence of debris flows were observed using periodic photographic images captured from 50–100 m above the ground level, using an unmanned aerial vehicle (UAV) (Table 1). We captured 605–1178 photographs to cover the entire debris flow channel in each photograph period. The point clouds of ground surface topography were constructed from the photographs captured by the UAV, using the structure from motion (SfM) analysis (Agisoft Metashape software). The spacing of cloud points ranged from 0.02–0.10 m. We used ground control points (GCPs) in the construction of points clouds, to reduce the errors, when the photographs were captured by UAVs that were not equipped with a real-time kinematic (RTK) system. The coordinate values in a JGD2000

rectangular coordinate system at 14 GCPs, which were different in all the periods, were positioned according to the static measurements obtained using global navigation satellite system (GNSS) devices (TOPCON, GRS-1) and the RTK measurements obtained using GNSS devices (Hemisphere, A101, A325, and R320). The number of GCPs positioned by the GNSS was

decreased to eight on August 11, 2021, because of the destruction of the points by debris flows. The stable boulders clearly identified in images (maximum 29 boulders), whose coordinate values were obtained using RTK-UAV (Phantom 4 RTK) photogrammetry, were also used as the GCPs. Digital elevation models (DEMs), with a grid size of 0.1 m, were also built using the Metashape software and a triangulated irregular network (TIN) model. The mean and standard deviations of the difference in the elevation in the stable areas between two consecutive DEMs were smaller than 0.1 and 0.3 m, respectively. The bedrock

topography beneath the channel deposits in the upper Ichinosawa was estimated from 1-m grid DEMs obtained using airborne Light Detection And Ranging (LiDAR) scanning and periodical UAV-SfM (Imaizumi et al., 2017, 2019). The lowest elevation of each grid cell among DEMs from 2005 was assumed to be the bedrock surface. The total volume of sediment storage in the initiation zone of debris flow (upper Ichinosawa above P20) was calculated from difference between DEMs obtained by periodic UAV-SfM (Table 1) and bedrock topography.


Table 1: Timing of unmanned aerial vehicle (UAV) photography

| Date of photography | UAV type | Grid size of DEMs (m) | Number of photographs | Debris flow timing after the photography |
|---|---|---|---|---|
| May 7, 2020 | Inspire 2 | 0.1 | 1065 | June 30–July 1, 2020 |
| July 3, 2020 | Phantom 4 Pro | 0.1 | 1178 | July 6–July 8, 2020 |
| September 6, 2020 | Phantom 4 Pro | 0.1 | 1123 | September 7, 2020 |
| September 14, 2020 | Phantom 4 RTK | 0.1 | 924 | March 21, 2021 |
| July 12, 2021 | Phantom 4 RTK | 0.1 | 606 | July 13, 2021 |
| August 11, 2021 | Phantom 4 Pro | 0.1 | 1090 | August 13–August 15, 2021 |
| August 24, 2021 | Phantom 4 RTK | 0.1 | 605 | May 18, 2022 |

### 3.6 Dimensionless numbers

In order to discuss the structure of force in the debris flows, Bagnold number ($N_B$), Savage number ($N_S$), and Friction number

($N_F$), which were proposed to describe debris flow regime, were calculated using the following equations (Iverson, 1997):

$$N_B = \frac{v_s \rho_s d^2 \gamma}{(1 - v_s)\mu} \qquad (5)$$

$$N_S = \frac{\rho_s d^2 \gamma^2}{(\rho_s - \rho_f)gh \tan \phi} \qquad (6)$$

$$N_F = \frac{v_s(\rho_s - \rho_f)gh \tan \phi}{(1 - v_s)\gamma\mu} \qquad (7)$$

where $v_s$ is the sediment concentration, $\rho_s$ is the particle density (2,650 kg m$^{-3}$), $\rho_f$ is the pore fluid density, $d$ is the particle

diameter, $\gamma$ is the flow shear rate (approximated to $uh^{-1}$, $u$ is the flow velocity), $\mu$ is the pore fluid viscosity, and $\phi$ is the internal friction angle (assumed to be 35°). In this study, $v_s$ was assumed to be 0.6, which is the typical value in steep channels of more than 20° (Takahashi, 1978; Lanzoni et al., 2017). The $\rho_f$ and $\mu$ were assumed to be 1200 kg m$^{-3}$ and 0.1 Pa s, respectively, which were measured values in Yakedake debris flow torrent, Japan (Takahashi, 1991). The $u$ in July 6, 2020 debris flow event was the front velocity of debris flow surges from P20 to P23 in TLC images. The $u$ in July 17, 2021 debris flow event was the

flow velocity in video images at P20, which is more accurate than the $u$ from TLC images. $d$ was set to 0.2 m, which is the median particle diameter of the channel deposits in the Ichinosawa catchment (Imaizumi et al., 2016a). Iverson (1997) indicated

the guidelines of interpretation of the dimensionless numbers based on experimental outcomes: collisional forces dominate over viscous forces for $N_B > 200$, collisional forces dominate over frictional forces for $N_S > 0.10$, and frictional forces dominate over viscous forces for $N_F > 2000$.

The dimensionless parameter $N_p$ evaluates the timescale ratio between the motion of a debris flow and the diffusion of disequilibrium pore fluid pressure as follows (Iverson and Denlinger, 2004):

$$N_p = \frac{\sqrt{Lg^{-1}}}{h^2 D^{-1}} \qquad (8)$$

where $L$ is the maximum length of the flow mass and $D$ is the diffusion coefficient. The excess pore-water pressure persists much longer than the time scale needed for a debris flow to flow into the downstream section when $N_p \ll 1$.

## 4 Results

### 4.1 Debris flow events

From April 2020 to November 2021, a total of 10 debris flows occurred in the study area (Table 2). The maximum 10-min rainfall intensity and total rainfall during all the debris flow events exceeded 24.0 mm h$^{-1}$ and 74.8 mm, respectively. The water pressure sensors recorded the debris flows at P20 during four rainfall events (on June 30 and July 6, 2020 and July 13 and
August 13, 2021). The debris flows on September 7, 2020 and May 21, 2021 terminated in the upper part of Ichinosawa and did not reach P20. The debris flow on March 21, 2021 occurred during the intermittent period of the monitoring. Additionally, the water pressure sensors did not record the pore-water pressure of the debris flows during three events (July 2, August 9, and August 17, 2021), due to mechanical issues.

**Table 2: Debris flows that occurred during the monitoring period**

| Date of rainfall events | Maximum 10-min rainfall intensity (mm h$^{-1}$) | Total rainfall depth (mm) | Volume of sediment storage before the event (m$^3$) | Volume of sediment storage after the event (m$^3$) | Monitoring of the pore-water pressure |
|---|---|---|---|---|---|
| June 30–July 1, 2020 | 49.2 | 309.2 | 45,431 | 50,438 | ○ |
| July 5–July 8, 2020 | 45.6 | 537.6 | 50,438 | 46,003 | ○ |
| September 7, 2020 | 50.4 | 193.0 | 46,003 | 40,692 | –[*3] |
| March 21, 2021 | 36.0 | 170.0 | – | – | –[*2] |
| May 21, 2021 | 24.0 | 211.0 | – | – | –[*3] |
| July 2–July 3, 2021 | 45.6 | 342.2 | – | 59,731 | –[*1] |
| July 13, 2021 | 140.4 | 74.8 | 59,731 | – | ○ |
| August 9, 2021 | –[*1] | 117.0 | – | 56,746 | –[*1] |
| August 13–August 15, 2021 | 54.0 | 473.0 | 56,746 | – | ○ |
| August 17–August 18, 2021 | 48.0 | 245.0 | – | 50,881 | –[*1] |

[*1] Mechanical issues in monitoring devices
[*2] Absence of water pressure sensors
[*3] Debris flow terminated before the P20 site

Among the four debris flows, for which we could observe the pore-water pressure, we could determine the topographic changes for three debris flows that occurred on June 30 and July 6, 2020 and July 13, 2021, using UAV-SfM. The topographic changes due to the debris flow that occurred on August 13, 2021 could not be determined using UAV-SfM, because the next debris flow (August 17–August 18, 2021) occurred before our UAV flight.

Periodical UAV surveys portrayed that the debris flow on June 30, 2020, was initiated around 1,850 m a.s.l. (near the summit of the basin) (Fig. 4a). However, the erosion of sediments was checked within 200 m from the initiation point, after which the debris flow began depositing sediments. The channel bed changes on the debris flow fan were less than 1 m (Fig. 4a). As we could not identify significant deposition of sediment below the P28 site, we conclude that the debris flow surges terminated around the P28 site.

The debris flow that occurred on July 6, 2020 significantly eroded the channel deposits (maximum depth of 6 m) above the P1 site. Sediment deposition was predominant in the sections from P3 to P11 and from P20 to P25. The deposition and erosion of sediments occurred repetitively in the section between P11 to P20, and the debris flow terminated around the P29 site below the site of small bed deformation (<2 m) in the section between P25 and P29.

The debris flow that occurred on July 13, 2021 eroded sediments from the initiation zone down to the P20 site. Thus, the length of the erosion zone was the longest among the three debris flow events. The deposition of sediment was significant around sites P22 and P23. Furthermore, small channel bed deformation continued to occur down to the lower end of the monitoring site (P36). The channel bed deformation could not be detected below the P36 site, due to the dense riparian forest cover in the region. The difference in the DEMs revealed that the volumes of sediments that reached the debris flow fan (below P20) during the debris flows that occurred on June 30 and July 6, 2020 and July 13, 2021 were 1,321, 1,905, and 1,374 $m^3$, respectively.

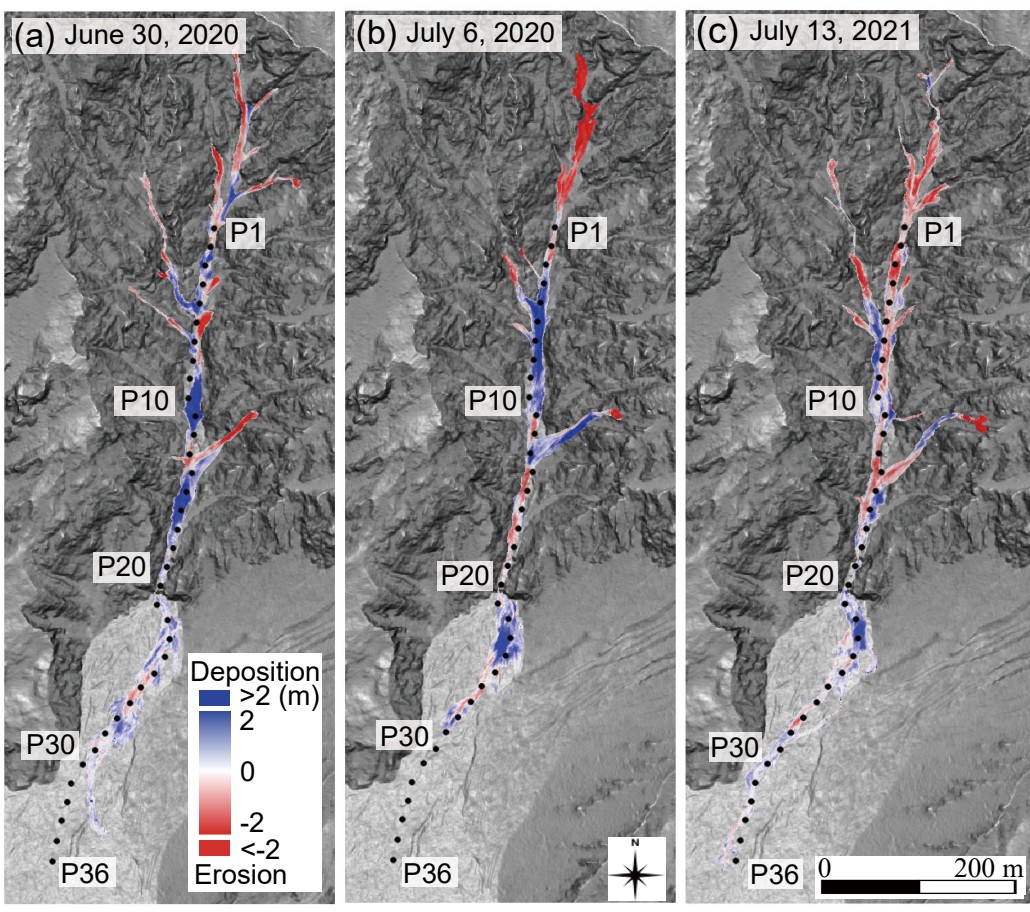

Figure 4. Changes in the topography of the study area caused by debris flows measured by the structure from motion technique using unmanned aerial vehicle (UAV-SfM). (a) Elevation change between May 7, 2020 and July 3, 2020 caused by the debris flow that occurred on June 30, 2020. (b) Elevation change between July 3, 2020 and September 6, 2020 caused by the debris flow that occurred on July 6, 2020. (c) Elevation change between July 12, 2021 and August 11, 2021 caused by the debris flow that occurred on July 13,

2021. Area of Fig. 4 is indicated by the black rectangle in Fig. 1. Background of the figures is the slope gradient of terrains calculated
from airborne light detection and ranging (LiDAR) digital elevation model (DEM) provided by Ministry of Land, Infrastructure,
Transport and Tourism (MLIT), Japan; dark and light grays indicate steep and gentle terrains, respectively.


## 4.2 Debris flow that occurred on June 30, 2020

A debris flow occurred just after the rainfall peak, on June 30, 2020, with a 10-min intensity of 49.2 mm h⁻¹ at 22:17 (hh:mm;

Fig. 5a). The water pressure sensors observed the changes in the pore-water pressure accompanying the runout of the debris flow

from 22:20 to 23:00. The flow types (partly and fully saturated) could not be captured by the TLCs, as the event occurred at

night. This debris flow was composed of at least six surges (Fig. 5b). The $\Delta p/\Delta z_{lower}$ of the first surge was similar or lower than

the depth gradient of hydrostatic pressure of clean water ($\partial p_w/\partial z$, 9,800 Pa m⁻¹; Fig. 5c). The peak $\Delta p/\Delta z_{lower}$ of the second

surge, which was highest during this debris flow event, was 47,273 Pa m⁻¹, which exceeded the depth gradient of normal stress

by the sediment (25,970 Pa m⁻¹). The $\Delta p/\Delta z_{lower}$ after the third surge was similar to the $\partial p_w/\partial z$, with the only exception being

the values calculated for the front of the surges.

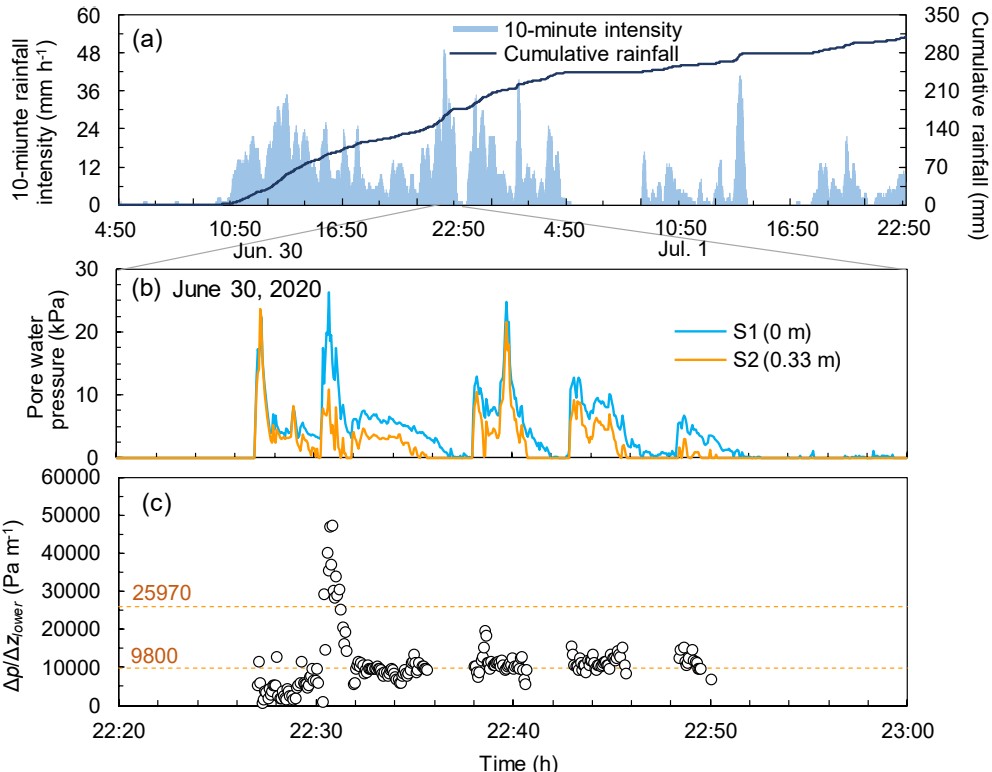

**Figure 5. Changes in the pore-water pressure during the debris flow that occurred on June 30, 2020: (a) cumulative rainfall and 10-min rainfall intensity of the entire event; (b) pore-water pressure; (c) gradient of pore-water pressure obtained from the monitoring**
**data from the S1 and S2 sensors**

## 4.3 Debris flow that occurred on July 6, 2020

On July 6, 2020, a debris flow was triggered by a rainfall peak (with a 10-min intensity of 45.6 mm h⁻¹, at 10:25) (Fig. 6a). The

two small fully-saturated surges arrived at P20 from 10:34. Then, a surge consisting of an unsaturated flow, followed by a

saturated flow (Fig. 6b), triggered the water pressure sensors. This surge deposited the sediments at P20, with a depth of about 50

cm, from 10:34 to 10:39. Hence, the monitored flow depth and pore-water pressure during this period were most likely affected

by the deposition of sediments. The pore-water pressure at the channel bed (S1 sensor) was similar to the hydrostatic pressure of

clean water in this period ($\rho gh$; Fig. 6c). Therefore, the $\Delta p/\Delta z_{entire}$ value in the first surge was similar to the $\partial p_w/\partial z$ value (9,800 Pa m$^{-1}$; Fig. 6d). The $\Delta p/\Delta z_{lower}$ in partly saturated part of this surge exceeded the $\partial p_w/\partial z$ value, while the $\Delta p/\Delta z_{lower}$ values in the subsequent fully saturated flow were similar to the $\partial p_w/\partial z$ values (Fig. 6c). A partly saturated debris-flow surge, which was the largest surge during this rainfall event, was monitored by a TLC at P20 (from 10:47). Notably, the pore-water pressure at the channel bed observed by the S1 sensor exceeded the hydrostatic pressure of clean water (Fig. 6c). The highest $\Delta p/\Delta z_{entire}$ values during this surge was 25,553 Pa m$^{-1}$. The $\Delta p/\Delta z_{lower}$ values in this surge highly exceeded the depth gradient of normal stress by the sediment (25,970 Pa m$^{-1}$) as well, indicating that the excess pore-water pressure occurred in the lower layer of the surge. Based on the TLCs data, we could conclude that the travel distance of this surge was the longest among all the surges that occurred during the rainfall event (Fig. 6b). A significant deposition of sediments in the section between P22 and P24 was most likely caused by this debris flow surge (Fig. 4b). Although the riverbed change occurred down to the P28 site (Fig. 4a), the TLCs could not capture the debris flow in the section from P24 to P28, because of the low flow height and the shade of topography. The surges after 10:49 were fully saturated because interstitial water could be identified on the flow surface in the images captured by the TLCs (Fig. 6b). The $\Delta p/\Delta z_{entire}$ and $\Delta p/\Delta z_{lower}$ values of these surges were similar to the $\partial p_w/\partial z$ value (Fig. 6d). Although the occurrence of debris flow surges continued until 15:25 in the upper part of the channel, no debris flow surge reached the P20 site after 10:52.

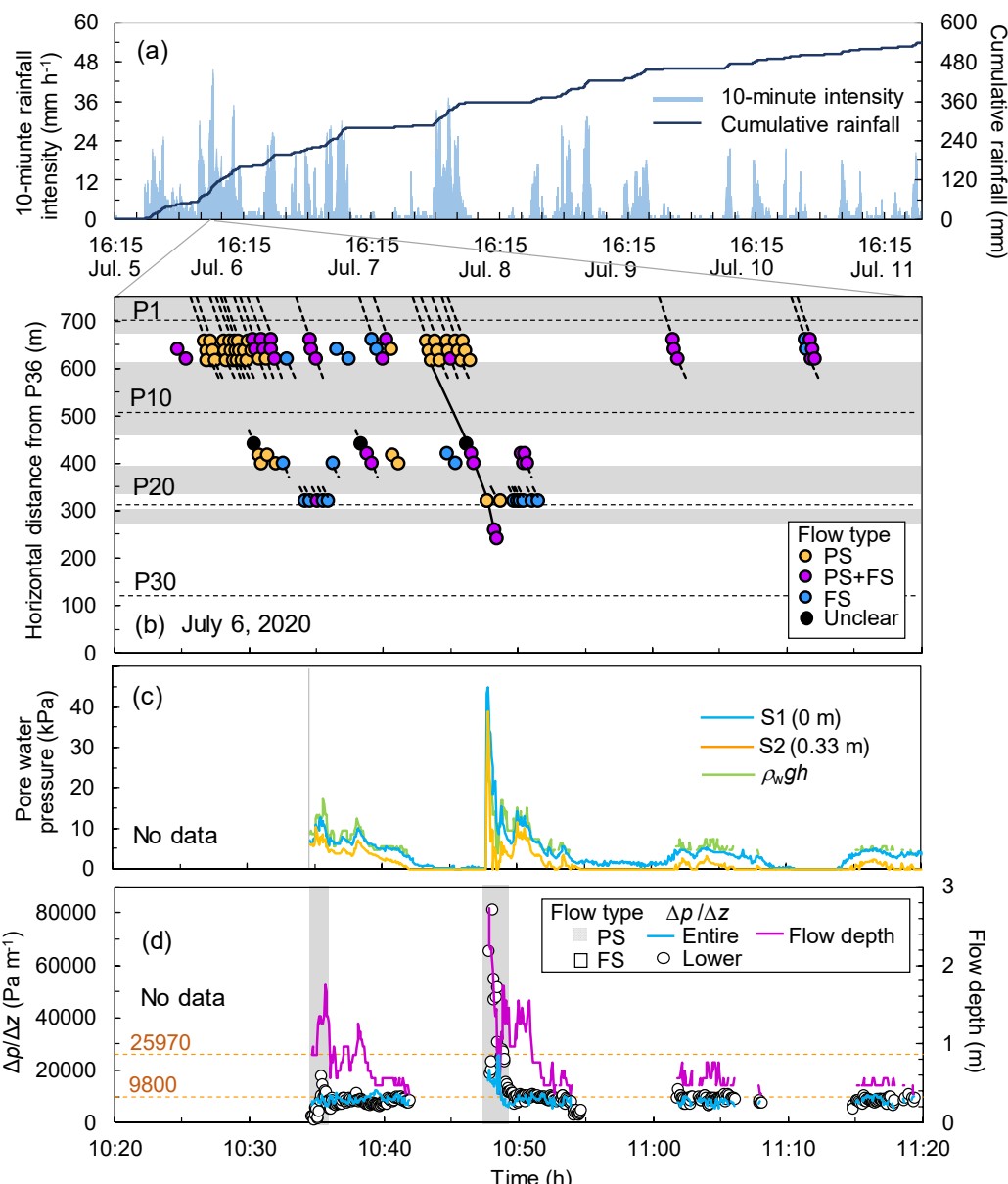

**Figure 6.** Changes in the pore-water pressure during the debris flow on July 6, 2020 (based on Imaizumi et al., 2023): (a) Cumulative rainfall and 10-min rainfall intensity of the entire event. (b) Migration of debris flow surges observed using time-lapse cameras (TLCs). PS and FS indicate partly and fully saturated flows, respectively. Flow type in gray sections could not be observed by the TLCs, because of hill shade of the topography. (c) Graph portraying pore-water pressure observed in the region; flow depth used for the calculation of hydrostatic pressure at the channel bed were obtained from TLC image analysis. (d) Gradient of pore-water pressure obtained from monitoring data acquired from the S1 and S2 sensors.

## 4.4 Debris flow that occurred on July 13, 2021

On July 13, 2021, a debris flow was triggered by a rainfall peak (with a maximum 10-min intensity of 140.4 mm h$^{-1}$) (Fig. 7a). The water pressures observed by S1 and S4 were used for the calculation of $\Delta p/\Delta z_{lower}$ during this event due to mechanical issues of S2 and S3. A total of 71 debris flow surges were identified using TLC images (Fig. 7b). Several surges were initiated in the section between P3 and P5, in which water was transported from tributaries to thick channel deposits (approximately 5 m

deep) in the main channel. Three debris flow initiation mechanisms were captured by the TLC images: erosion by overland flow proceeding over unsaturated deposits, sliding of channel deposits, and sediment supply from channel bank, talus slope, and tributaries. The predominant flow type was different in different channel sections; surges composed of partly saturated flow

dominated the section covered by thick channel deposits (between P1 and P16 and between P21 and P35), while most of the surges were composed of both fully and partly saturated flows in the sections that had exposures of the bedrock on the channel bed (between P17 and P20; Fig. 7b). A total of 10 debris flow surges passed the P20 site, reaching the alluvial fan below the site. The travel distance of this debris flow was longer than the debris flows that occurred on June 30, 2020 and July 6, 2021 (Fig. 4); three surges passed the lower end of the monitoring section at the P36 site (Fig. 7b). The largest surge with the flow depth of 3.4

m arrived at P20 at 15:09. Although this surge triggered the pressure sensors, the pore-water pressure was not observed during the surge, due to the start-up time of the sensors. The monitoring of the pore-water pressure started from 15:10, when a saturated flow was passing the P20 site (Video supplement 1). The pore-water pressure at the channel bed was much higher than the hydrostatic pressure of the clean water, during the passage of this saturated flow (Fig. 7c). Additionally, the $\Delta p/\Delta z_{entire}$ value exceeded the depth gradient of normal stress by the sediment (25,970 Pa m$^{-1}$). The water pressure sensors at three different

depths (S1, S4, and S5) observed the pore-water pressure in the debris flow surge from 15:11 (Fig. 7c, Video supplement 2). The TLC images portrayed that this surge was composed of an unsaturated flow, followed by a short fully-saturated flow. The $\Delta p/\Delta z_{lower}$ values of this surge were the same or lower than the $\partial p_w/\partial z$ values (9,800 Pa m$^{-1}$; Fig. 7d). This surge terminated at the P21 site, about 20 m downstream of the monitoring site P20. Only lowest sensor (S1 sensor) observed the pore-water pressure from the end of this surge to 15:14, as the flows were unsaturated and had low flow depths. During the passage of fully

saturated flow and at the tail of surges, the pore-water pressure at the channel bed (S1 sensor) exceeded the hydrostatic pressure of clean water ($\rho g h$). The $\Delta p/\Delta z_{entire}$ values portrayed fluctuations, but there were no significant changes in the flow depth. Then, a partly saturated surge, with a peak flow depth of 1.76 m, was observed at the P20 site, from 15:14:10 (hh:mm:ss). The pore-water pressure observed by the S1 sensor was much smaller than the hydrostatic pressure of clean water, resulting in $\Delta p/\Delta z_{entire}$ values that were lower than the $\partial p_w/\partial z$ values. Additionally, the S4 sensor did not detect any increase in the pore-

water pressure. This surge passed the lower end of the monitoring section at the P36 site. Another partly saturated surge reached the P20 site at 15:14:35. However, we could not obtain the data from this surge, due to the significant sediment deposition at the P20 site.

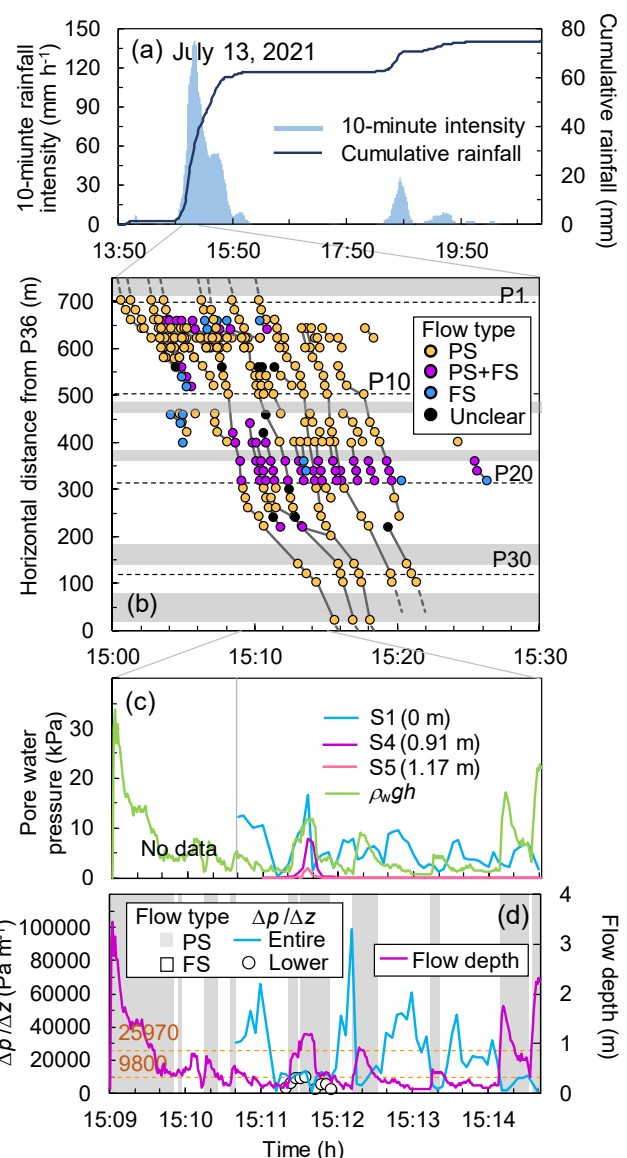

**Figure 7. Changes in the pore-water pressure during the debris flow on July 13, 2021 (based on Imaizumi et al., 2023): (a) Cumulative rainfall and 10-min rainfall intensity of the entire event. (b) Migration of debris flow surges observed by time-lapse camera (TLC) images. PS and FS indicate partly and fully saturated flows, respectively. Flow type in the gray sections could not be observed by TLC images, because of the shade of the topography. (c) Graph portraying the pore-water pressure observed in the region; the flow depths used for the calculation of hydrostatic pressure at the channel bed were obtained from TLC image analysis. (d) Gradient of pore-water pressure obtained from the monitoring data acquired of S1 and S4 sensors.**

## 4.5 Debris flow that occurred on August 13, 2021

On August 13, 2022, a debris flow occurred just after the rainfall peak (with the 10-min intensity of 54.0 mm h$^{-1}$; at 20:20) (Fig. 8a). The monitoring process of the water pressure was initiated by the signal from the rope sensor at 20:58. the fluctuations in the pore-water pressure and flow depth were observed during the passage of the first surge. The different flow types (partly and fully saturated) could not be captured by the TLC images, as these events occurred at night. The $\Delta p/\Delta z_{lower}$ values of the first surge was similar or lower than the $\partial p_w/\partial z$ values (9,800 Pa m$^{-1}$; Fig. 8c). The pore-water pressure monitored by the S1 sensor was in the range of 3,000–3,500 Pa after the first peak, affected by the deposition of sediments over the S1 sensor. As the pore-water

pressure was similar to $\rho_w gh$ (Fig. 8b), the flows during this period were most likely fully saturated, with the weight per unit volume being 1,000 kg m⁻³. The peak of the pore-water pressure during the second surge (monitored from 21:54) was clearer than that during the first surge. Both the $\Delta p/\Delta z_{entire}$ and $\Delta p/\Delta z_{lower}$ values fluctuated around the value of $\partial p_w/\partial z$ (9,800 Pa m⁻¹) after the peak of the second surge.

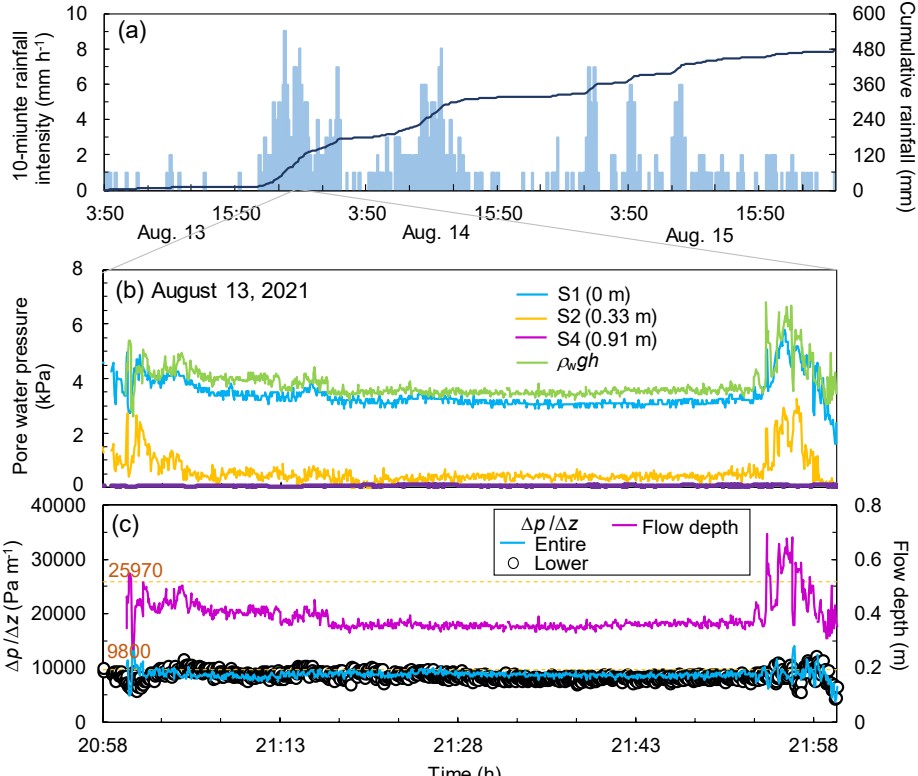

**Figure 8: Changes in the pore-water pressure during the debris flow that occurred on August 13, 2021. (a) Cumulative rainfall and 10-min rainfall intensity of the entire event; the precipitation data was provided by the Ministry of Land, Infrastructure, Transport and Tourism. (b) Migration of debris flow surges; PS and FS indicate partly and fully saturated flows, respectively. (b) Graph portraying the pore-water pressure observed in the region; the flow depth used for the calculation of hydrostatic pressure at the channel bed were obtained from a laser distance sensor. (c) Gradient of pore-water pressure obtained from the monitoring data acquired from the S1 and S4 sensors**

### 4.6 Depth gradient of pore-water pressure

To determine the factors that affecting the magnitude of $\Delta p/\Delta z$, we calculated the time average of $\Delta p/\Delta z$ during the passage of the main flow of each surge (Table 3). Both the $\Delta p/\Delta z_{entire}$ and $\Delta p/\Delta z_{lower}$ values ranged from 8,000 Pa m⁻¹ to 12,000 Pa m⁻¹, similar to the values of $\Delta p_w/\Delta z$ (9,800 Pa m⁻¹), while some surges highly exceeded or were below the range. The $\Delta p/\Delta z_{lower}$ value exceeded $\Delta p/\Delta z_{entire}$ value in most surges. Although both the $\Delta p/\Delta z_{entire}$ and $\Delta p/\Delta z_{lower}$ values during the debris flow that occurred on July 6, 2020 were the highest among all the monitored debris flows, their maximum 10-min rainfall intensity and cumulative rainfall depth were lower than those that occurred on July 13, 2021 and June 20, 2020, respectively (Table 3). The $\Delta p/\Delta z_{entire}$ values did not have a clear relationship with the flow depth (Fig. 9a). However, the $\Delta p/\Delta z_{entire}$ values of the debris flow surge monitored from 10:47:46 (hh:mm:ss) on July 6, 2020, which had the highest flow depth, was the highest among all the monitored surges. The $\Delta p/\Delta z_{lower}$ values of the debris flow surge monitored from 10:47:46 on July 6, 2020 was also the highest among all the debris flow surges, although the $\Delta p/\Delta z_{lower}$ values did not have a clear relationship with the pore-water pressure at the channel bed (Fig. 9b). The maximum 10-min rainfall intensity on July 13, 2021 (140.4 mm h⁻¹) was the highest among all the monitored debris flow events. However, the $\Delta p/\Delta z_{lower}$ values of the debris flow surge during this event

was not higher than the other events (Fig. 9c). Clear relationship was not identified between the cumulative rainfall and $\Delta p/\Delta z_{lower}$ values (Fig. 9d).

The Bagnold numbers ($N_B$) of the monitored debris flow surges exceeded 200, indicating that the collisional forces dominated over the viscous forces (Fig. 10a). The savage numbers ($N_S$) of the partly saturated surges on July 6, 2020 and the surges monitored from 10:47:46 on July 13, 2021 were below 0.10; thus, the frictional forces dominated the collisional forces (Fig. 10b). The $N_S$ of the surge monitored from 10:48:38 on July 13, 2021 was larger than 0.10; hence, the collisional force dominated the frictional force. $N_S$ of the surge monitored from 10:49:46 on July 13, 2021 was almost 0.10. The friction numbers ($N_F$) of the monitored surges were above 2000, indicating that the friction forces dominated over the viscous forces (Fig. 10c). The P number ($N_P$) of the monitored surges were much smaller than 1; thus, generated excess pore-water pressure persisted over long terms (Fig. 10d). The surges with the highest value of $\Delta p/\Delta z_{entire}$ had the smallest values of $N_B$, $N_S$, and $N_P$ and the largest value of $N_F$.

**Table 3. Depth gradient of pore-water pressure in debris flow surges. Time average of the gradient during the passage of main flow has been listed in the table. The surges affected by sediment deposition over the channel bed have been excluded in the table.**

| Date | Period of main flow | | Maximum 10-min rainfall (mm h⁻¹) | Cumulative rainfall at the rainfall peak (mm) | Maximum flow depth (m) | Average flow depth (m) | $\frac{\Delta p}{\Delta z}_{entire}$ (Pa m⁻¹) | $\frac{\Delta p}{\Delta z}_{lower}$ (Pa m⁻¹) | Predominant flow type |
|------|-------|-----|-----|-----|-----|-----|-----|-----|-----|
| | Start | End | | | | | | | |
| June 30, 2020 | 22:27:04 | 22:27:36 | 49.2 | 163.8 | - | | - | 4,091 | - |
| | 22:30:24 | 22:31:16 | 49.2 | 163.8 | - | | - | 30,487 | - |
| | 22:38:04 | 22:38:32 | 49.2 | 163.8 | - | | - | 10,511 | - |
| | 22:39:28 | 22:40:16 | 49.2 | 163.8 | - | | - | 10,385 | - |
| | 22:43:00 | 22:43:32 | 49.2 | 163.8 | - | | - | 11,162 | - |
| | 22:48:24 | 22:48:52 | 49.2 | 163.8 | - | | - | 12,424 | - |
| July, 6, 2020 | 10:47:46 | 10:48:30 | 45.6 | 99.6 | 2.72 | 1.73 | 16,158 | 43,909 | Partly saturated |
| | 10:48:38 | 10:49:14 | 45.6 | 99.6 | 1.75 | 1.34 | 8,150 | 20,636 | Partly saturated |
| | 10:49:46 | 10:51:36 | 45.6 | 99.6 | 1.56 | 1.28 | 8,726 | 9,833 | Fully saturated |
| July 13, 2021 | 15:11:24 | 15:11:43 | 140.4 | 33.0 | 1.20 | 0.96 | 9,314 | 8,473 | Partly saturated |
| | 15:12:15 | 15:12:28 | 140.4 | 33.0 | 0.92 | 0.72 | 7,901 | -* | Partly saturated |
| | 15:14:10 | 15:14:34 | 140.4 | 33.0 | 1.76 | 0.95 | 6,312 | -* | Partly saturated |

*No calculation could be carried out because the upper sensors did not detect pore-water pressure

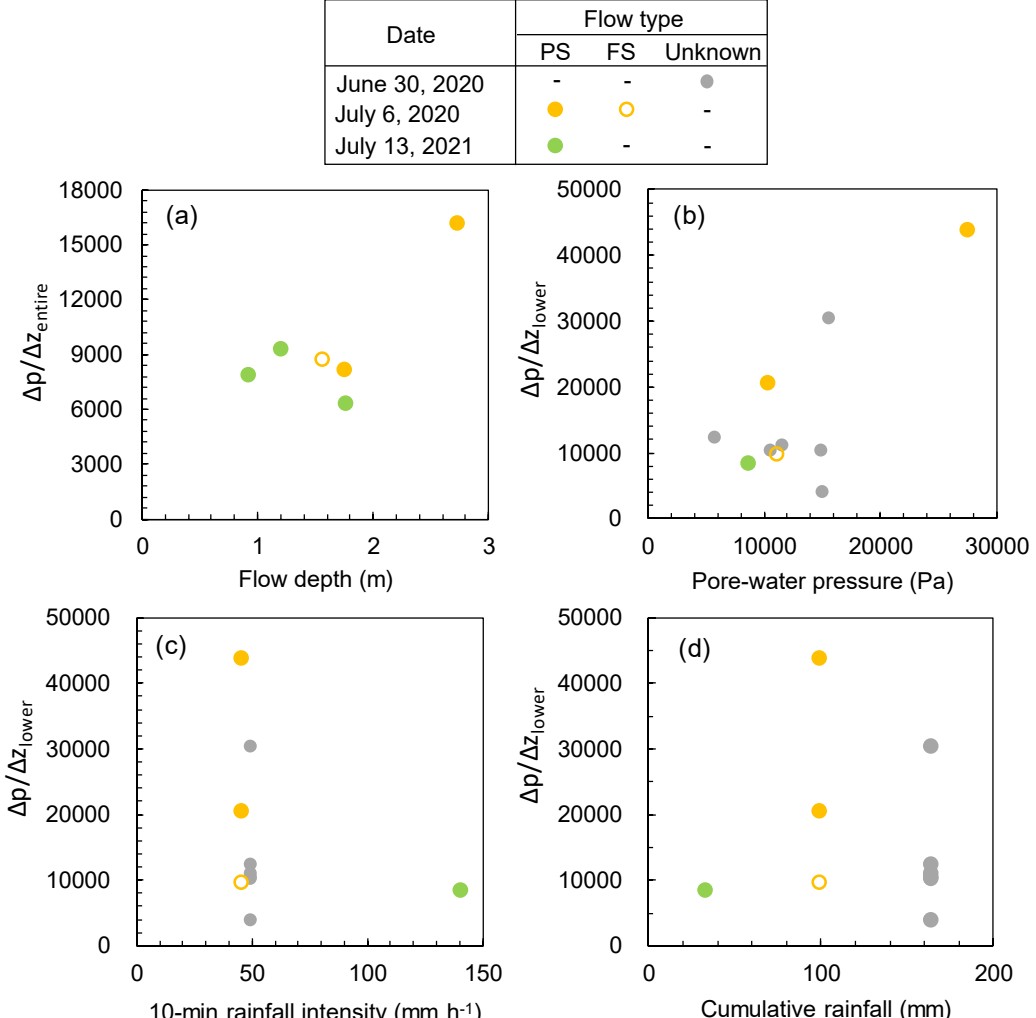

**Figure 9.** Depth gradient of pore-water pressure ($\Delta p/\Delta z$) during a discharge peak compared with the flow magnitude and rainfall characteristics: (a) the flow depth and depth gradient of pre-water pressure of the entire flow, (b) pore-water pressure at channel bed (S1) and depth gradient of pore-water pressure in the lower part of flow, (c) the 10-min rainfall intensity preceding the surge and depth gradient of pore-water pressure in the lower part of flow, and (d) cumulative rainfall and depth gradient of pore-water pressure in the lower part of flow. Note that the height of the higher sensor [$h_u$ in Eq. (4)] during the debris flow event on July 13, 2021 was different from the other events. The surges observed in the periods with sediment cover on the channel bed were excluded from the comparison.

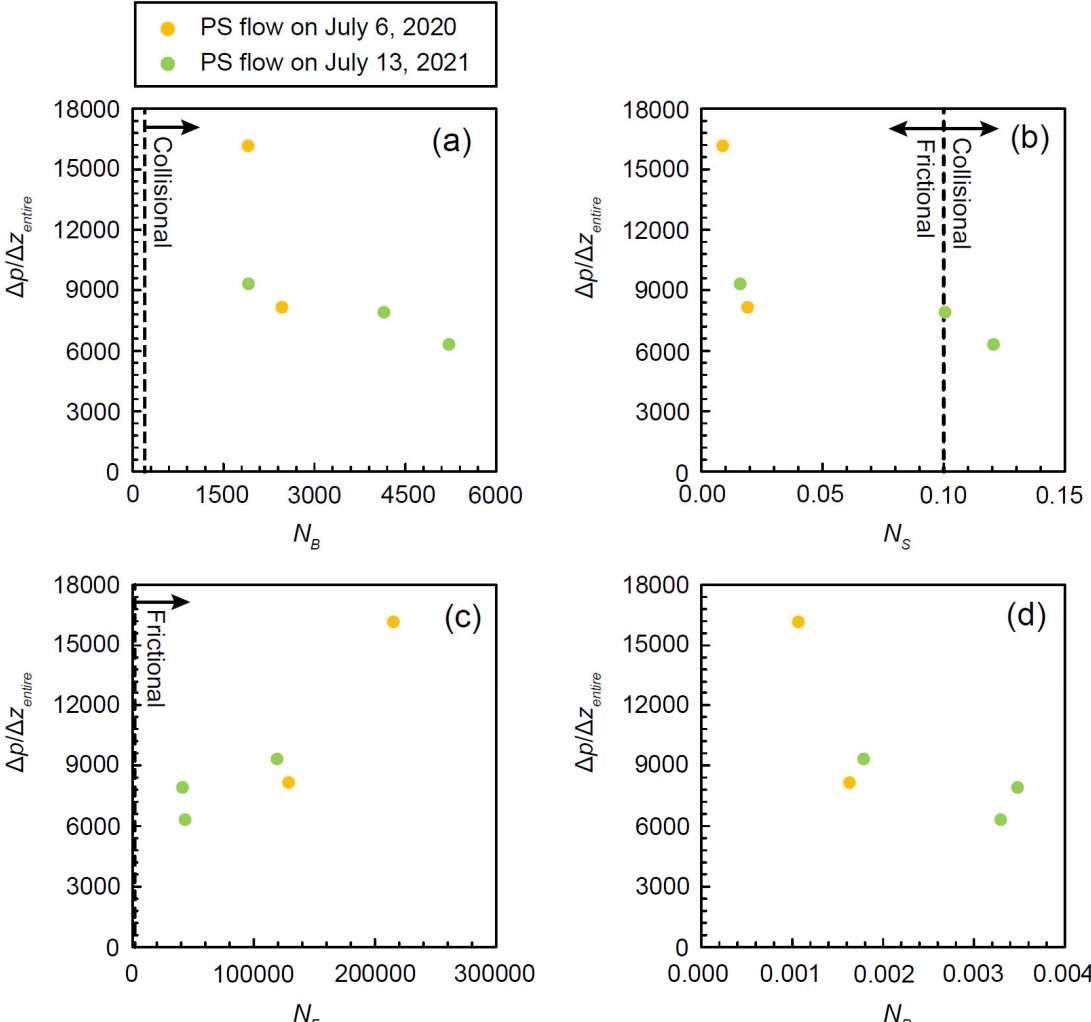

**Figure 10. Comparison between the dimensionless numbers and depth gradient of pore-water pressure of the entire flow ($\Delta p/\Delta z_{entire}$) during a discharge peak: (a) Bagnold number ($N_B$), (b) Savage number ($N_S$), (c) Friction number ($N_F$), and (d) P number ($N_P$). The dashed lines indicate the boundary between collisional, frictional, and viscous flow regimes (Iverson 1997).**

## 5 Discussion

### 5.1 Pore-water pressure in debris flow

Previous field studies monitored the excess pore-water pressure in the fully saturated debris flows (Berti et al., 2000; McArdell et al., 2007; McCoy et al., 2010; Nagl et al., 2020), while the data of the excess pore-water pressure in partly saturated debris flow was limited to laboratory experiments (Okada and Ochiai, 2008). Our study revealed that the pore-water pressure can exceed the hydrostatic pressure of clean water ($\rho g h$) in both fully and partly saturated debris flows (Figs. 6, 7). However, the pore-water

pressure in another debris flow was similar to the hydrostatic pressure of clean water (Fig. 8). The $\Delta p/\Delta z_{entire}$ values reported for other torrents, obtained from the pore-water pressure at channel beds divided by the flow depth, was in the range of 4900–19600 Pa m$^{-1}$ (Berti et al., 2000; McArdell et al., 2007; McCoy et al., 2010; Nagl et al., 2020). The time averaged $\Delta p/\Delta z_{entire}$ value of all the debris flow surges considered in this study was in the range of the pore-water pressure (Table 3). However, the $\Delta p/\Delta z_{lower}$ values calculated for many of surges exceeded the range, indicating that lower part of the flow had a higher depth

gradient of pore-water pressure than the upper part of the flow. The gradient of the hydrostatic pressure ($\partial p_h / \partial z$) in the upper part of the partly saturated debris flow was low, due to the unsaturation of the flow (Imaizumi et al., 2005). Additionally, because the $\Delta p/\Delta z_{lower}$ values of some surges exceeded the depth gradient of the normal stress by the sediment (25,970 Pa m$^{-1}$; Figs. 5–8; Table 3), it was important to consider excess pore-water pressure ($p_e$ in Eq. (1)), to explain the depth gradient of the pore-water pressure. Another potential factor affecting difference between $\Delta p/\Delta z_{entire}$ and $\Delta p/\Delta z_{lower}$ is the accuracy of their estimation. Because $\Delta p/\Delta z_{lower}$ focuses on a smaller spatial scale, the $\Delta p/\Delta z_{lower}$ can represent the depth gradient of the pore-water pressure in a specific section more precisely than the $\Delta p/\Delta z_{entire}$.

The $p_s$ in Eq. (1) is controlled by the suspension of fine sediments that has been discussed using the settling velocity in previous studies (Dietrich, 1982; Jiménez and Madsen, 2003). Rouse number, which is a non-dimensional number in fluid dynamics used to define a suspension of sediment, is expressed by the following equations (Rubey,1933; Cheng and Chiew, 1999; Sakai et al., 2019):

$$P = \frac{w_0}{100\kappa u_*} \qquad (9)$$

$$w_0 = \sqrt{sgd}\left[\sqrt{\frac{2}{3}+\frac{36\nu^2}{sgd^3}} - \sqrt{\frac{36\nu^2}{sgd^3}}\right] \qquad (10)$$

where $P$ is the Rouse number, $w_0$ is the settling velocity (cm), $\kappa$ is the von Kármán constant (0.41), $u_*$ is the friction velocity ($=\sqrt{gRI}$), $R$ is the hydraulic radius (m), $I$ is the energy gradient, $d$ is the particle size (cm), $\nu$ is the coefficient of kinematic viscosity (cm$^2$ s$^{-1}$), $s$ is the weight per unit volume of sediment in water ($= \rho_s\rho_f^{-1} - 1$). The Rouse number determining the threshold condition of suspension in debris flow (ranging from 0.116 to 0.813) is smaller than the threshold of suspended sediment in fluvial channels due to the shading effect by boulders inside of the flow (Nishiguchi, 2014; Sakai et al., 2019). By assuming that $R$ was the average flow height of monitored surges (1.07 m; Table 3), $I$ was the local channel gradient (=sin 16°), $\rho_s$ = 2,650 (kg m$^{-3}$), $\rho_f$ = 1,000 (kg m$^{-3}$), and $\nu$ = 0.01 (cm$^2$ s$^{-1}$), the particle sizes that provide $P$=0.11 and 0.813 were 0.07 and 3.0 cm, respectively. Rate of sediment particles finer than 3.0 cm in the channel deposits of Ichinosawa catchment, which was analyzed by sieving and in-situ measurement (Imaizumi et al., 2016a), was about 18 %, implying that materials of the suspended sediment existed in the channel deposits. Although $\rho$ and $\nu$ values were possibly affected by the suspension of fine sediments, the pore-water pressure coming from suspended particles ($p_s$ in Eq. (1)) likely affects magnitude of the pore-water pressure. At the same time, because the pore-water pressure in some debris flow surges was almost same as the hydrostatic pressure (Figs. 5, 6, 8), magnitude of the $p_s$ was low when the volume of erosible fine sediments on the channel bed surface was small.

In partly saturated debris flows, viscous force was less important than frictional and collisional forces (Figs. 10a, 10c), affected by higher ratio of gravels in debris flow material (Imaizumi et al., 2016a) and the poor interstitial fluid. Scale of the pore space between boulders due to the high volumetric solid concentration, resulting in the predominance of the frictional force (Figs. 10b, 10c). Additionally, the $\Delta p/\Delta z_{lower}$ was higher in the partly saturated debris flows with larger influence of the frictional force (Figs. 10b, 10c). Consequently, boulders highly affect magnitude of pore-water pressure in the partly saturated debris flow. A partly saturated flow that occurred on July 6, 2020, had the highest $\Delta p/\Delta z_{lower}$ value, among all the monitored debris flows (80,909 Pa m$^{-1}$). The peak $\Delta p/\Delta z_{lower}$ value of the June 30, 2020 event was also high (47,273 Pa m$^{-1}$). Although we could not classify the flow type (fully and partly saturated) for the debris flows that occurred on June 30, 2020, due to the absence of TLC images, the deposition of sediments in the steep channel sections above P1 (>30°) implied the occurrence of partly saturated debris flows (Fig. 3; Imaizumi et al., 2017). The $\Delta p/\Delta z_{lower}$ values of these events exceeded the depth gradient of the normal stress by the sediment (25,970 Pa m$^{-1}$), indicating that excess pore-water pressure surely existed in the lower part of these partly

saturated flows (Figs. 5 and 6). The excess pore-water pressure occurs due to the contraction of interstitial water by shearing of grain-fluid assembly (Iverson, 1997, 2005; Iverson and George, 2014; Kaitna et al., 2016), Reynolds stress from the turbulence of interstitial fluid caused by the collision of boulders (Zenit and Hunt, 1998; Hotta and Ohta, 2000; Hotta, 2011), and centrifugal force in the curved channel section (Hotta, 2012). The contraction of interstitial water affected the high depth gradient of the pore-water pressure in partly saturated flows, because the depth gradient in the lower layer ($\Delta p/\Delta z_{lower}$), in which the shear stress was higher than that in the upper layer, was higher than the $\Delta p/\Delta z_{entire}$ value. The largest magnitude of the $\Delta p/\Delta z_{lower}$ and $\Delta p/\Delta z_{entire}$ values during the surge (monitored from 10:47:46 on July, 6, 2020), corresponding to the highest flow height and pore-water pressure at the channel bed (Fig. 9), also indicated that the contraction of interstitial water affected the generation of excess pore-water pressure. Laboratory experiments and physical analysis reveal that the excess pore-water pressure is high in flows that have a layer containing large particles in the upper part of the flow (Yang et al., 2021a, 2021b). The structure of the partly saturated flow, with the upper unsaturated layer being mainly composed of cobbles and boulders, contributed to the high excess pore-water pressure in the debris flows that occurred in the Ichinosawa catchment. The excess pore-water pressure is maintained over long timescales when the hydraulic diffusivity in the debris flow is low (Iverson et al., 2004; Kaitna et al., 2016). $N_p$ values of the monitored debris flow surges were much smaller than 1 (Fig. 10d), indicating that these surges sustain the excess pore-water pressure over long terms once the pressure is generated (Iverson, 1997). The debris flow surges with the highest $\Delta p/\Delta z_{entire}$ value had the lowest $N_p$ value, suggesting that the low hydraulic diffusivity causes the high pore-water pressure gradient.

Dynamic pore-water pressure in Bernoulli's principle is expressed by the following equation:

$$p_d = \frac{\rho_f v^2}{2} \qquad (11)$$

where $p_d$ is the dynamic pore-water pressure other than Reynolds stress, $v$ is the flow velocity. The dynamic pore-water pressure $p_d$ need to be considered in Eq. (1) when the flow velocity along a specific direction continuously exists. In our case, because pressure sensors were installed on a subvertical bedrock (Fig. 3), $v$ in Eq. (11) is the lateral flow velocity rather than the vertical flow velocity. Our pressure sensors were covered by the mortar to reduce the direct hitting of the lateral flow to sensors. Additionally, the depth gradient of pore-water pressure was similar to hydrostatic pressure even if the flow depth exceeded 1 m (Fig. 6). Therefore, although we do not have any data on the lateral velocity of the interstitial fluid, impact of $p_d$ on the observed pore-water pressure is likely small.

Furthermore, on July 13, 2021, we observed high $\Delta p/\Delta z_{entire}$ values in fully saturated flows from 15:12 (hh:mm) to 15:14 (Fig. 7c). As the entrainment of boulders was not identified in the TLC images, we could conclude that the Reynolds stress and suspension of fine sediments increased the pore-water pressure of the saturated flows that occurred in this period (Zenit and Hunt, 1998; Hotta and Ohta, 2000; Hotta, 2011).

Kean and Staley (2011) reported that the timing of the peak excess pore-water pressure was synchronous to the peak 10-min rainfall intensity. This implies that short rainfall intensity affects the magnitude of the pore-water pressure. Rainfall intensity and cumulative rainfall affect the water content of unconsolidated materials both on hillslopes and channels (Kean and Staley, 2011; Hürlimann et al., 2015). The water content in the debris flow material potentially affects depth gradient of the pore-water pressure by controlling the sediment concentration in the debris flow. However, relationships between rainfall factors and the depth gradient of pore-water pressure was not clear in our monitoring results (Table 3, Figs. 9c, 9d). The $\Delta p/\Delta z_{lower}$ and $\Delta p/\Delta z_{entire}$ values in the partly saturated flows that occurred on July 13, 2021, were similar or lower than the $\partial p_w/\partial z$ values, whereas the rainfall intensity of this event was the highest amongst all the four debris flow events considered in this study. Hence,

the existence of an unsaturated layer in the debris flow may obscure the relationship between the rainfall intensity and magnitude of excess pore-water pressure.

**5.2 Debris flow mobility**

In previous studies, hydrostatic pressure was used to estimate the equilibrium sediment concentration and boundary of the fully and partly saturated layers in debris flows (Takahashi, 1978, 2014; Imaizumi et al., 2017). The critical channel gradient that
separates the partly and fully saturated flows, from the aspect of static force, can be determined using the following equation (Imaizumi et al., 2017):

$$\tan\alpha = \frac{(1-n)(\gamma_s - \gamma_f)}{(1-n)\gamma_s + n\gamma_f}\tan\phi \qquad (12)$$

where $\alpha$ is the channel gradient, $n$ is the porosity, $\gamma_s$ is the force of gravity acting on a unit volume of sediment, $\gamma_f$ is the force of gravity acting on a unit volume of interstitial fluid for debris flow), and $\phi$ is the effective internal angle of friction. In the
540 Ichinosawa catchment, by assuming that $\phi = 37.3°$, $n = 0.3$, $\gamma_s = 26,000$ (kg m$^{-2}$ s$^{-1}$), and $\gamma_f = 9,800$ (kg m$^{-2}$ s$^{-1}$), we calculated that $\alpha = 22.2°$ (Imaizumi et al., 2017). However, the runout of partly saturated debris flow surges was monitored for the channel gradient of 15–20° (Figs., 1d and 7b). For example, a partly saturated debris flow surge observed by the monitoring system at the P20 site from 15:14 passed the lower end of the monitoring site (P36) that had a channel gradient of 15–18°. The runout of partly saturated debris flows in gentler channel sections were also monitored in other torrents (McArdell et al., 2007; Okano et al.,
2012; McCoy et al., 2013). A pore-water pressure that is higher than the hydrostatic pressure of clean water increases the mobility of debris flow (Iverson and Vallance, 2001; McArdell et al., 2007; McCoy et al., 2010; Tang et al., 2018; Zhou et al., 2018), resulting in the runout of unsaturated flow in gentler channel sections. Note that the pore-water pressure in this debris flow surge was lower than hydrostatic pressure at P20 (Fig. 7c). Significant deposition of sediments by this debris flow was identified at apex of the debris flow fan by the periodical UAV-SfM (Fig. 4c). This deposition possibly decreased ratio
of boulders in the flow, changing the stress and pressure structures in the flow. Thus, the pore-water pressure become higher, resulting in the long travel distance of the debris surge.

Although the pore-water pressure in the debris flow that occurred on July 6, 2020 was higher than the flows that occurred on other dates, the former's travel distance was much shorter. Therefore, factors other than the pore-water pressure also affect the mobility of debris flows. The volume of the debris flow material in the initiation zone of a debris flow, which changes with time
due to sediment supply and transport processes, also controls the characteristics of the flow (Bennett et al., 2013; Gregoretti et al., 2016; Imaizumi et al., 2017; Rengers et al., 2020). However, the effect of the volume of the debris flow material on the debris flow mobility was not significant in this study, because the total volume of the debris flow material was similar among all the monitored debris flow events (Table 2). Takayama et al. (2022) revealed that the infiltration of interstitial water of debris flows into unsaturated channel deposits can decrease the mobility of debris flows. The debris flow that occurred on July 13, 2021,
which had the longest travel distance amongst all the three debris flows observed using the UAV, was triggered by an extremely intense rainfall peak (Table 2, Fig. 8). Intense rainfall increases the water contents of the surficial layer of the channel deposits on the debris flow fan, possibly decreasing the infiltration of interstitial water into the channel. In contrast, the short distance of the debris flow that occurred on July 6, 2020 may have been affected by the infiltration of interstitial water into the channel deposits on the debris flow fan, due to lower rainfall intensity.

## 6 Summary and conclusion

In this study, we monitored fully and partly saturated debris flows in Ohya landslide scar, central Japan, using water pressure sensors at multiple depths, TLCs, and a video camera. We could successfully obtain the data on the pore-water pressure in the fully and partly saturated flows during four debris flow events. The depth gradient of the pore-water pressure was different among all the debris flow events and surges. The depth gradient of the pore-water pressure in the lower part of the flow was generally higher than that in the upper part of the flow. The pore-water pressure at the channel bed of some partly saturated flows was higher than the hydrostatic pressure of clean water, even though the upper layer was unsaturated. The contraction of the interstitial water generated excess pore-water pressure, resulting in a high pore-water pressure. The depth gradient of the fully saturated debris flows was similar to the hydrostatic pressure of clean water, while some saturated flows portrayed higher pore pressure than the hydrostatic pressure, possibly due to Reynolds stress and the suspension of fine particles. The travel distance of the debris flows, which were investigated by periodical UAV-SfM (structure from motion using unmanned aerial vehicle) and the monitoring of TLCs, was long during a rainfall event with high intensity, although the pore-water pressure in the flow was not significantly high. Therefore, we could conclude that the mobility of debris flow was controlled not only by the pore-water pressure, but also by other factors, such as the moisture level of the channel deposits.

Our study revealed that high pore-water pressure enables partly saturated debris flows to travel in the gentler channel sections. Notably, the flow type (fully and partly saturated) should be considered, to estimate the pore-water pressure in debris flows. Although our study presented the characteristics of pore-water pressure in partly saturated debris flow, the generation mechanism of high pore-water pressure is unclear. Therefore, laboratory experiments, physical modelling, and further field monitoring must be promoted, to understand the fluid mechanisms in partly saturated debris flows. As many debris flows occur in steep channel sections that partly saturated debris flow can occurred, the understanding of pore-water pressure in such steep channels is essential for improving the estimations and predictions of the timings and magnitudes of debris flows.

*Data availability*

Corresponding author has all data analyzed in this study. Most of data can be shared upon agreement of authors. For further information, please contact to the corresponding author.

*Supplement*

The video supplement 1 is available on-line at https://doi.org/10.5446/62074. The video supplement 2 is available on-line at https://doi.org/10.5446/62072.

*Author contributions*

SO designed the study. SO, FI designed and installed field monitoring system. All of the authors conducted field motoring. SO and FI analysed monitoring data. ST contributed discussion on the physical mechanism of debris flow. SO and FI contributed to the writing of the paper.

*Competing interests*

The authors declare that they have no conflict of interest

*Acknowledgements*

We are grateful to Tomohiro Egusa for his advice on analysis of the paper. UAV survey was conducted with the support by
Yuichi Hayakawa. We deeply appreciate Hideaki Takahashi and other students in Shizuoka University for their help of field
survey.

*Financial support*

This study was funded by Ministry of Land, Infrastructure, Transport and Tourism, Japan. This study was also supported by
by JSPS Grant Numbers 21K18790 and 90596438.

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
