# Peer review of "Field monitoring of pore-water pressure in fully and partly saturated debris flows at Ohya landslide scar, Japan"

_EGUsphere, 2023_

## Author Response (AR1)

**Reviewer 1**

We sincerely thank you for the efforts you have made to improve our submission to "*Earth Surface Dynamics*". Comments from the reviewer are very helpful for us to improve our manuscript. We could deepen the discussion by the comment about the flow velocity of surges. We have responded to all review comments in the following paragraphs. Labels and line numbers after our response correspond to those in the revised manuscript with tracked changes.

[Comment R1_1] In this study, the authors present that debris flows exhibit different characteristics during rainfall events, even within the same gully, and the internal stress of debris flow interacts with these variations. The data obtained in this study are highly valuable, including detailed measurements of pore-water pressure, as well as diagrams illustrating the propagation of debris-flow surges with high spatiotemporal resolution in Figs 6 and 7. These findings provide useful insights for predicting debris flow runout and therefore make a significant contribution to the relevant research fields. However, the manuscript requires a few improvements, particularly in the interpretation of the measured excess pore-water pressure.

[Reply] It is our pleasure that the manuscript was acclaimed by the reviewer.

[Comment R1_2] Major concerns:
Clarification of excess pore water pressure:
   The assumption of a potential maximum pore-water pressure of 25970 Pa/m, considering fully suspended sediment within the pore space of debris flow, seems reasonable (L170-171). However, it is important to note that this value also represents the theoretical maximum value for the normal stress in debris flows. The authors should discuss why the measured pore-water pressure in some cases (Figs. 5 & 6) exceeded this theoretical maximum value. It is possible that the pressure gauge detected lateral "dynamic" pressure, which could be influenced by the agitation of sediment particles, despite the authors' aim to measure the "static" pressure using the pressure gauges buried in the sidewall. If this is the case, separating the static excess pore-water pressure from the measured values would be challenging. The authors should provide a justification or at least discuss this discrepancy.

[Reply] Based on the comment from anther reviewer, we have revised terminology of this value. Now we call this as "depth gradient of normal stress by the sediment" (e.g., line 315). As the reviewer pointed out, the 25970 Pa/m is the theoretical maximum value for the normal static pressure in interstitial fluid. We have clarified it in the text

(Lines 180-181).

As described in the Fig. 3, sensor was completely covered by mortal, preventing direct hit of the lateral flow component. Additionally, inlet of the water into the sensor faces downward direction. Therefore, impact of lateral dynamic pressure on the measured value is considered to be small. We have revised explanation of the structure of water-pressure sensors (lines 156-158). Additionally, our monitoring results also show that the depth gradient of pore-water pressure is similar to hydrostatic pressure even if the flow depth exceed 1 m (Fig. 6), indicating that influence of lateral dynamic pressure is not very large. At the same time, we agree that effect of the dynamic pressure on the observed pore-water pressure need to be discussed. We have discussed impact of dynamic pressure in the discussion section (line 526-535).

In order to clarify generation mechanisms of pore-water pressure, we have evaluated effect of suspension of sediments on the increasing in pore-water pressure (lines 475-492). Additionally, we have calculated Bagnold number, Savage number, and Friction number, and Np value (new Fig. 10, lines 495-500, 522-525). We hope these discussions contribute understanding of pore-water pressure in the debris flow.

[Comment R1_3] Insufficient analyses in section 4.6:

   The authors compare dp/dz with flow depth and pore-water pressure on an average-value basis. However, utilizing temporal dp/dz would enable a more detailed and effective comparison by classifying flow conditions (FS, PS) based on the TLS data, in accordance with the classification in Fig. 2b. Additionally, considering the discussion on mobility in section 5.2, dp/dz could be compared with other effective indices besides flow depth and pore-water pressure (Fig. 9). The authors have data on the propagation of each debris-flow surge (Fig. 5b and Fig. 6b), which allows for the determination of debris-flow velocity. By assuming local steady-state conditions and a certain sediment concentration, for example, the friction coefficient could be calculated to evaluate the energy dissipation of the target surges. This type of analysis would enhance/enrich the discussion.

[Reply] We have calculated dp/dz in each debris flow surges (Table 3, Fig. 9, Fig. 10). We have compared dp/dz with dimensionless parameters (Bagnold number, Savage number, and Friction number), which were calculated using the monitored flow velocity (Fig. 10, lines 495-500). We have focused this analysis in partly saturated debris flows because estimated flow velocity of fully saturated flow likely has a larger error because of less clear changes in the discharge (e.g. Fig. 6c). We hope these new analyses have deepened discussion in our study.

Specific minor comments:

[Comment R1_4] L109 and L568: Change "2016a" to "2016."
[Reply] We have revised the published year of article. Because we have added the other paper "Imaizumi et al., 2016a", the citation is currently Imaizumi et al., 2016b (lines 112 and 689).

[Comment R1_5] eq. (1), L171-172: What is the causal factor of the excess pore water pressure?
[Reply] Previous studies suggested that the excess pore-water pressure occurs due to the contraction of interstitial water by the surrounding boulders in a high shear stress environment, Reynolds stress from the turbulence of interstitial water caused by the collision of boulders, and centrifugal force in curved channel sections. We have briefly introduced the causal factors after this sentence (lines 170-172). Additionally, as explained above, we have deepened discussion on the causal factor of the excess pore-water pressure and hydraulic diffusivity of the flow by calculating dimensionless parameters and Np value (lines 492-497, 522-526).

[Comment R1_6] L179: S1 and S2 are indicated as the set of two sensors, but is S4 also used to calculate dp/dz(lower) in Fig. 7?
[Reply] In Fig. 7, sensors S1 and S4 were used to calculate dp/dz(lower). We have clarified (lines 191-195, 355-356).

[Comment R1_7] eq. (4): Should "pu-pl" be changed to "pl-pu"?
[Reply] As pointed by the reviewer, "pu-pl" should be negative value if the numerator is "pu-pl". We have revised the expression (lines 191-194).

[Comment R1_8] L231: Refer to Table 2 at this point (Table 2 is not mentioned in section 4.1).
[Reply] We have referred Table 2 in this sentence (line 266).

[Comment R1_9] L287: "...triggered by a..."
[Reply] We have revised the sentence (line 322).

[Comment R1_10] L296 (L381, 382): Change "10:47:46" to "10:47"?
[Reply] We will remove the second as suggested (line 331).

[Comment R1_11] L306 (L433, 455): Remove "h" from the time "10:52 h."
[Reply] We will remove h (lines 341, 515, 561).

[Comment R1_12] L438-443: Can the influence of rainfall be discussed in the same context as the suspension of fine sediment and Reynolds stress? The relationship between rainfall and pore water pressure is unclear, while the suspension and Reynolds stress reflect the internal stress of debris flow.
[Reply] We have added a new figure showing the relationship between the rainfall intensity and pore-water pressure (Fig. 9c). We could find a clear relationship because rainfall intensity was not different so much among debris flow surges. We also compared the cumulative rainfall with the depth gradient of the pore water pressure (Fig. 9d), but we could not find a clear relationship. Currently we do not have an idea on direct effect of the rainfall characteristics on the pore-water pressure. We have added discussion on the relationship between the rainfall and pore-water pressure (lines 541-545).

[Comment R1_13] L453-460: The debris-flow surges exemplified here were partly or fully saturated at P20 (Fig. 7b), and the pore-water pressure was identical or less than hydrostatic (Fig. 7d, although the data are limited to a few surges). Do the authors consider that these debris-flow surges gained excess pressure in the downstream section where the flows would have been in the deposition phase? If so, what causal factor increased the pore-water pressure?
[Reply] We feel sorry that we cannot directly answer to this question because we observed the pore-water pressure just at P20. Although we are also interested in the pore-water pressure on the debris flow fan, it was impossible to set up monitoring devices because of significant aggradation and degradation of the channel bed and frequent changes in the debris flow path (avulsion). Potential cause is deposition of sediments just below P20 (Fig. 4c). This decreased coarse particles in the flow, and increased mobility of flow. We have added discussion why the travel distance was long during event (lines 566-569).

[Comment R1_14] L562: Is this reference to Imaizumi et al. (2022)?
[Reply] This was published in proceedings in 2023. We have revised (lines 345, 684).

[Comment R1_15] L569-: Add line breaks for references to Imaizumi et al. (2019) and Itoh et al. (2019).
[Reply] We have added line breaks (lines 690-697).

[Comment R1_16] L615: Place McArdell et al. (2007) in L607.

[Reply] We will change the order of references as pointed out by the reviewer (lines 735-736).

Thank you again for reviewing our manuscript.

**Reviewer 2**

We sincerely thank you for the efforts you have made to review our manuscript. Comments from the reviewer, such as comments on the rainfall, are very helpful for us to improve the manuscript. We have responded to all review comments in the following paragraphs. Labels and line numbers after our response correspond to those in the revised manuscript with tracked changes.

[Comment R2_1] General Comments
This manuscript describes an innovative field measurement program aimed to investigate pore water pressure in fully and partially saturated debris flows. The authors have done a very nice job preparing this manuscript. It is clearly written, the figures are well prepared and clear, and the logic is easy to follow. Consequently, I have very few comments overall because this manuscript is in good shape already.
[Reply] It is our pleasure that the manuscript was acclaimed by the reviewer.

[Comment R2_2] First, I commend the authors on such a detailed study. The type of data they present is very difficult to obtain, and consequently very valuable to the community in general. I have one suggestion on how rainfall intensity is presented (same comment shown below in the specific comments). It looks like when they discuss the 10 minute rainfall intensity they are showing the total rainfall (mm) in a 10 min period. However, in hydrology, traditionally people use units of mm/hr for intensity. So a 10 minute intensity would be mm/10min, converted to units of mm/hr. I would suggest changing this throughout the paper.
[Reply] As the reviewer comments, our intensity in the rainfall depth in a 10- min period. We have converted the unit to mm/hr in the text, table, figures (Table 2, Figs 5 to 9)

[Comment R2_3] Second, I think at the end of the abstract, they slightly undersell the big conclusions. I think the authors could concentrate on the importance of pore water pressure in the largest surges. The authors could also emphasize the uniqueness of this dataset.
[Reply] Thank you for your suggestion. We have revised the last part of the sentence in the abstract as suggested by the reviewer (lines 25–28).

[Comment R2_4] Finally, I'm also curious if you could see if there is a relationship between the peak 10 min intensity preceeding a surge and the peak dpdz? Not sure if

there is a relationship there, but it might be worth looking at.

[Reply] We have added a new figure showing comparison of the 10 min intensity to $dp/dz_{lower}$ (Fig. 9c). We did not show peak $dp/dz_{lower}$ but time average of $dp/dz_{lower}$, because it was difficult to determine peak value due to the fluctuation of the value during a debris flow surge. The relationship between the rainfall intensity and $dp/dz_{lower}$ was not clear. We also compared cumulative rainfall and $dp/dz_{lower}$, but the relationship was not clear (Fig. 9d). We have added discussion between rainfall factors pore-water pressure (lines 541-545).

[Comment R2_5] Specific Comments:
23-24: I'm not sure if this is the 'main point' with which you want to end the abstract because, realistically, I'm not sure that the flow type is something that can really be "considered" ahead of time. Maybe you could be more specific, and say it needs to be considered in modeling. I think a stronger finding from your paper is that you found that excess pore water pressure was present in many of the biggest surges, and that it appears to be an important mechanism in debris flow surge behavior.

[Reply] As suggested by the reviewer, we have emphasized importance of findings in this study at the end of abstract (lines 25-28).

[Comment R2_6] 29: change "decrease" to "decreases"
[Reply] We have revised as pointed out by the reviewer (line 32).

[Comment R2_7] 107: Can you label the Hontani torrent on the map?
[Reply] We have labeled the Hontani in Fig. 1.

[Comment R2_8] 108: You mention P36, but you haven't told us what P means yet.
[Reply] We have deleted the statement "P36". Because Hontani has been labeled in the Fig. 1, readers can understand location of the junction from the map now (lines 110-111).

[Comment R2_9] 121: change "intermitted" to "intermittent"
[Reply] We have replaced "intermitted" to "intermittent" (line 125).

[Comment R2_10] 123: Is "intermission season" Nov. to March? Can you specify this more clearly?
[Reply] This is from November to March. We have clarified (line 127).

[Comment R2_11] 136: reference the supplement where readers can see this video.
[Reply] Link to the video supplement has been described in line 609. We have clarified that the video information can be seen at the end of the article (line 140).

[Comment R2_12] 153: Change "accumulations was" to "accumulations were"
[Reply] We have revised the sentence (line 159).

[Comment R2_13] 232: Here and elsewhere you it looks like when you discuss the 10 minute rainfall intensity you are showing the total rainfall (mm) in a 10 min period. However, in hydrology, traditionally people use units of mm/hr for intensity. So a 10 minute intensity would be mm/10min, converted to units of mm/hr. I would suggest changing this throughout the paper.

[Reply] We have revised the unit throughout the text as suggested (e.g., line 309). We also converted units in figures and tables (Figs. 5 to 9, Table 3)

[Comment R2_14] 276: If possible, can you indicate that 22:20 to 23:00 represents the time of day in hh:mm.
[Reply] We have add explanation of hh:mm and hh:mm:ss in the manuscript (lines 310, 380, 419, 536).

[Comment R2_15] 416: after "flows" can you add in the highest dp/dzlower value in ()?
[Reply] We think the sentence was in line 414. We have added the value in the sentence (line 499).

[Comment R2_16] 464: you indicate that the volume of material depends on the initiation zone, but there have been a few studies that indicate that the volume of material supplied to channels declines during parts of the year when debris flows are active, and then refills during winter periods. I would suggest considering adding references to these, because that helps to explain your declining volume over time: Rengers, F. K., Kean, J. W., Reitman, N. G., Smith, J. B., Coe, J. A., & McGuire, L. A. (2020). The influence of frost weathering on debris flow sediment supply in an alpine basin. Journal of Geophysical Research: Earth Surface, 125, e2019JF005369. https://doi.org/10.1029/2019JF005369
Bennett, G., Molnar, P., McArdell, B., Schlunegger, F., & Burlando, P. (2013). Patterns and controls of sediment production, transfer and yield in the Illgraben.

Geomorphology, 188, 68–82.

[Reply] Thank you for your suggestion. We have added references suggested by the reviewer (lines 572-574).

[Comment R2_17] 492: Change "thigh" to "high"

[Reply] We have revised the "high" (line 602).

[Comment R2_18] Figure 1. Say what "P1 means"

[Reply] We have added explanation on P1 to P36 in the figure caption (Fig. 1).

[Comment R2_19] Figure 3a. Can you use an arrow to show the flow direction?

[Reply] We have added an arrow showing flow direction in Fig. 3a (Fig. 3a).

[Comment R2_20] Figure 4. I'm about to suggest something that is 100% a personal preference. I know there are different approaches to blue/red plots of erosion and deposition. I prefer erosion = red and deposition = blue. You may not agree with this, and that's totally acceptable, but consider switching to that color scheme.

[Reply] Based on the reviewer's comment, we have checked color of erosion/deposition areas in previous papers. We could find both red/blue and blue/red patterns. Papers using red for the erosion would be major. We have changed the color (Fig. 4).

[Comment R2_21] Table 2: Can you show the volume of sediment after the event?

[Reply] We have added sediment volume after the events (Table 2). We have added detail of UAV flight after the event (Table 1).

Thank you again for reviewing our manuscript.

**Reviewer 3**

We sincerely thank you for the efforts you have made to review our manuscript.
Comments from the reviewer are very helpful for us to deepen our discussion on the
physical mechanism of debris flow. We have responded to all review comments in the
following paragraphs. Labels and line numbers after our response correspond to those in
the revised manuscript with tracked changes.

Major Concerns:

[Comment R3_1] The author's explanations for "excess pore pressure" and the "high
depth gradient of the pore-water pressure" are not sufficiently comprehensive, and the
cited reasons lack rigor.
First, regarding the loading of particles, there is a lack of reliable evidence to suggest
that this pore-water pressure is induced by the loading of particles. The load of particles
becomes part of the pore-water pressure only when particles are fully suspended by
fluid, which usually requires a condition typically associated with smaller particle sizes.
As indicated in Figure 2, partly saturated debris flows often occur at the head of debris
flows where larger particles accumulate. In such conditions, the movement of particles
and fluid is markedly different, and the loading of particles does not contribute
significantly to the rise in pore-water pressure. Determining whether particles are fully
suspended often relies on parameters such as the Rouse number (Rouse, 1937; Dietrich,
1982; Jiménez & Madsen, 2003) and Stokes number (e.g., Ancey et al., 1999). This
necessitates knowledge of the mean or median grain size of the debris flow and slurry
viscosity, among other parameters. The shear rate can be roughly estimated from
surface flow velocity and flow depth. If slurry viscosity has not been measured, it can
be estimated based on the typical viscosity range observed in field studies of debris
flows in Japan or other regions, such as Illgraben, Chalk Cliff, and Jiangjia Ravine.

Thank you so much for your comments that deepen physical analysis of the pore water
pressure. We think our terminology was not very clear in the previous version of the
manuscript. We have revised terminology, such as loading of particles. We agree that
the suspension of sediments increases $p_s$ value in Eq. (1). Based on the comment from
the reviewer, we have added analysis of the Rouse number (lines 475-494). Our analysis
implied that material of suspended particles existed in the debris flow material. At the
same time, because the pore-water pressure in some debris flow surges was similar to

hydrostatic pressure of clean water, effect of the $p_s$ on the pore-water pressure was likely small when fine sediments were poor on the surface of the channel deposits.

 [Comment R3_2] Secondly, in addressing the "contraction of interstitial water," it is crucial to analyze the diffusion time of pore pressure, which describes the tendency for pore fluid pressure to develop between moving grains. This can be assessed through parameters such as the Darcy number (Iverson, 1997; Lanzoni et al., 2017) and P number (Iverson et al., 2004), which evaluate the timescale ratio between avalanche motion and the diffusion of disequilibrium pore fluid pressure. Contraction of interstitial water leading to "excess pore pressure" is more likely to occur when this timescale ratio is relatively small. However, these conditions are stringent in natural debris flow, typically requiring finer particle sizes, higher solid-phase concentrations, and greater interstitial fluid viscosity.

 We have added analysis on the P number. The excess-pore water pressure can be maintained over longer timescales because the diffusion of the pore-fluid pressure was not high in the Ichinosawa (lines 259-263, new Fig. 10d, 522-526). We hope analysis of the P number supports our idea that the contraction of interstitial water generated excess pore-water pressure.

 [Comment R3_3] "Excess pore pressure" and the "high depth gradient of the pore-water pressure" are central issues in this study. Therefore, when discussing and analyzing these phenomena, the authors should provide quantitative analysis (if feasible, estimates of missing parameters can be approximated to facilitate final calculations) or detailed qualitative discussions, followed by explanations of excess pore pressure, rather than presenting vague concepts.

Based on the comments from the reviewer, we have calculated Bagnold number, Savage number, and Friction number. New analyses revealed that frictional force dominated in the partly saturated debris flows (Figs. 10a to 10c). Scale of the pore space between boulders due to the high volumetric solid concentration, resulting in the predominance of the frictional force. Additionally, the $\Delta p/\Delta z_{lower}$ was higher in the partly saturated debris flows with larger influence of the frictional force. We have added discussion on the internal force affecting the pore-water pressure (lines 495-500). We also added related references in the paper.

Minor comments:

[Comment R3_4] Figure 1: The figure displays only three observation points, yet they are labeled from P1 to P20 to P36. If there are additional observation points between P1 and P36 that are relevant and need to be shown, it is advisable to include them on a separate, larger-scale map. If only these three points are pertinent to the study, renumbering is suggested.

Because it is difficult to show all of analysis points in the Figure 1, we have displayed their locations in Figure 4. We have clarified it in the caption of Figure 1.

[Comment R3_5] Lines 170-175: The statement "the maximum weight per unit volume of the interstitial water was the same as that of the sediments" is not in accordance with reality, as every liquid has its maximum sediment transport capacity.

We also think that this value does not have reality as maximum weight per unit volume of the interstitial water. This value is not lower boundary of occurrence of the excess pore-water pressure. But the excess pore-water pressure surely (clearly) exists in debris flows when the pore-water pressure exceeds this value. We have improved explanation and terminology of this value throughout the paper (lines 177-183).

[Comment R3_6] Lines 175-180: The two methods for calculating the gradient of pore-water pressure (Eq. 3 and Eq. 4) do not differ fundamentally; they mainly vary in the size of the delta_h interval. Since Eq. 4 offers a smaller interval and higher precision, the method relying on bulk flow depth estimation, as represented by Eq. 3, may be less accurate as it overlooks local information within the intermediate flow layer.

We think that the spatial difference in the gradient of the pore-water pressure along the depth direction affects disagreement of Eq. 3 and Eq. 4 values. At the same time, as pointed out by the reviewer, difference in the accuracy of estimation also affects disagreements of the two values. We have explained potential effect of the estimation accuracy on the difference in the values from Eq. 3 and Eq. 4 (lines 472-474).

[Comment R3_7] Figure 5: What is the unit of rainfall intensity? Is it expressed in mm/day or mm/hour? Although the authors mention "the 10-minute rainfall intensity," this is not a commonly used unit for representing rainfall intensity. Clarification is needed.

We have changed unit of the intensity to mm h$^{-1}$ throughout the manuscript (e.g., Table 2, Figures 5 to 9).

[Comment R3_8] Line 492: There is a spelling error in this section.

We have revised misspelling in line 492 (line 602 in new version).

Thank you again for your review.